# Scaling the Scaling Logic: Agentic Meta-Synthesis of Logic Reasoning

**Bowen Liu** [1 2]  **Zhi Wu** [1]  **Runquan Xie** [1]  **Zhanhui Kang** [1]  **Jia Li** [2]

## Abstract

Reinforcement Learning from Verifiable Rewards (RLVR) is bottlenecked by data: existing synthesis pipelines rely on expert-written code or fixed templates, confining growth to instance-level perturbations. We shift the evolvable unit from problem instances to task-family specifications. SSLogic is an agentic meta-synthesis framework in which LLM agents iteratively author and refine executable Generator–Validator pairs inside a closed Generate–Validate–Refine loop, producing families with new rules and difficulty gradients rather than parameter variations of old ones. A Multi-Gate Validation Protocol—multi-strategy consensus plus Adversarial Blind Review, where independent agents solve each instance by writing and executing code—filters ill-posed tasks before they enter training. Starting from 400 seed families, two evolution rounds yield 953 families and 21,389 verifiable instances. Three converging comparisons (step-matched, token-matched, and size-controlled on external Enigmata data) consistently show higher training utility of evolved data, with gains of SynLogic +5.2, AIME25 +3.0, and BBH +5.5 on Enigmata. Fine-grained KORBench evaluation reveals selective improvements in logic (+13.2%) and operation (+9.6%), linking structural evolution to downstream gains. Code is available at https://github.com/AdAstraAbyssoque/Scaling-the-Scaling-Logic.

## 1. Introduction

Large Reasoning models, exemplified by o1 and DeepSeek-R1, have demonstrated significant scalability across a wide range of complex tasks (OpenAI et al., 2024; DeepSeek-AI et al., 2025a; Comanici et al., 2025). As model scale and

[1]Hunyuan, Tencent [2]The Hong Kong University of Science and Technology (Guangzhou). Correspondence to: Jia Li <jialee@ust.hk>.

*Proceedings of the $43^{rd}$ International Conference on Machine Learning*, Seoul, South Korea. PMLR 306, 2026. Copyright 2026 by the author(s).

training paradigms evolve, further enhancement of reasoning capabilities increasingly relies on stable, verifiable, and scalable training signals. Reinforcement Learning (RL) and its application in verifiable reward settings have been proven to yield critical gains (Schulman et al., 2017; Shao et al., 2024; Hu et al., 2024), yet training efficiency is highly constrained by the supply of large-scale, low-noise supervision and feedback.

Among various verifiable training signals, logical reasoning tasks possess unique advantages: their semantics and constraints can be formally expressed, answers are programmatically verifiable, and they can be organized as task families (Liu et al., 2025b). This enables explicit control over structural variations and difficulty gradients at the task level, providing verifiable training signals with lower noise. We further view logical tasks as a testbed for general reasoning primitives: by minimizing reliance on domain-specific knowledge, they isolate core challenges such as symbolic manipulation, constraint propagation, and multi-step deduction, making them highly suitable for learning from intermediate reasoning steps and optimizing reasoning trajectories under verifiable rewards. Existing research has shown positive results in small-scale (mostly less than $10^2$ task families) logical task synthesis and training (Liu et al., 2025; Chen et al., 2025a; Stojanovski et al., 2025). Scaling to larger, sustained regimes still requires minimally supervised synthesis of high-quality, verifiable, and evolvable task families, which remains a key bottleneck.

To mitigate this data bottleneck, the community has proposed various logical data synthesis frameworks, as summarized in Figure 1. One line of work (Tafjord et al., 2021; Liu et al., 2025; Chen et al., 2025a; Stojanovski et al., 2025) primarily relies on expert-written code scripts to scale task production; another (Morishita et al., 2024; Helff et al., 2025; Yu et al., 2025; He et al., 2025) extends instances within established skeletons like ILP/SAT/PDDL. Although these methods effectively increase data scale, most remain confined to parameter augmentation and surface perturbation within fixed templates or inference graphs, essentially constituting *Instance Synthesis*. This paradigm struggles to transcend the priors of predefined structures, thereby hindering the move towards task-family-level evolution.

Addressing these limitations, we propose **Scaling the Scal-**

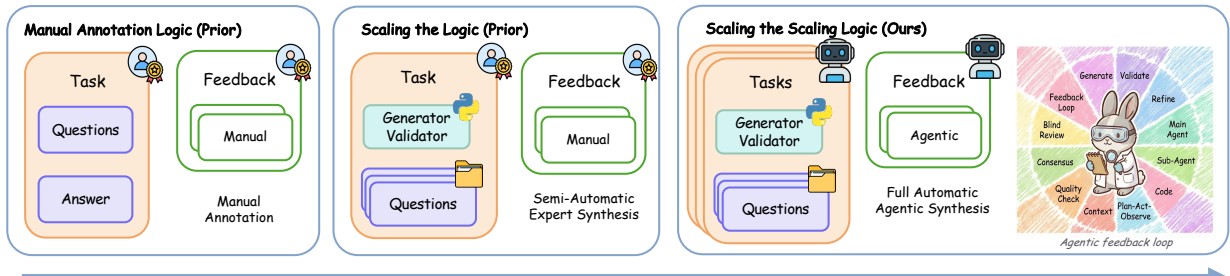

*Figure 1.* Paradigm Shifts in Logic Data Generation: From Manual Curation to Agentic Meta-Synthesis. **Left:** Traditional Manual Curation focuses on Task/QA pairs, where quality control and feedback rely heavily on humans. **Middle:** Code Synthesis introduces executable Generators/Validators, achieving partial automation but still requiring manual supervision. **Right:** Our Agentic Meta-Synthesis enables fully automatic, end-to-end data production. Agents iteratively generate and validate task families (Generator + Validator) and instances, realizing the path from Manual → Semi-Automatic → Full-Automatic construction (Scaling the Scaling Logic).

**ing Logic (SSLogic)**: an *Agentic Meta-Synthesis* framework achieving automatic, continuous expansion of logical task families at the code level. Unlike approaches that only generate problem text or parameters, SSLogic employs agents as program synthesis and refine engines, iteratively updating Executable Specifications in a Generate–Validate–Refine closed loop. Consequently, the task family specification—defining generation and verification rules—becomes an object that can be searched, refined, and extrapolated. This shifts the evolvable object from *problem instances* to *task family specifications*, enabling the continuous production of new families and rules while maintaining verifiability and controllable difficulty.

To address quality and validation challenges in synthetic data, we design a *Multi-Gate Validation Protocol*. On one hand, we perform consistency checks on the same instance by integrating multi-strategy validators to reduce systematic biases from single implementations. On the other hand, we introduce independent agents for Adversarial Blind Review, forcing them to solve problems by writing and executing code, thereby strictly filtering out ambiguous descriptions, ill-posed instances, and implicit logical loopholes.

Starting from 400 seed task families spanning seven major categories (Appendix E), through two rounds of iteration, we expanded the task families from 400 to 953 and verifiable instances from 5,718 to 21,389. Experiments demonstrate that data evolved via SSLogic exhibits higher value in downstream RL training, yielding not only improvements on SynLogic (+5.2) and BBEH (+1.4), but also cross-domain gains on AIME25 (+3.0) and Brumo25 (+3.7). Beyond performance metrics, we analyze code-level and algorithmic conceptual changes during evolution, shedding light on how synthesized data drives improvements in LLM reasoning.

## 2. Preliminaries

We formalize our setting as verifiable task-family synthesis and introduce core quantities for analysis.

### 2.1. Problem Setting

**Task families.** We model logical reasoning tasks as *verifiable task families*. A task family is a tuple $\mathcal{T} = (G, V)$, where the *generator* $G$ samples instances $\boldsymbol{x} = (\boldsymbol{x}_{\text{text}}, \boldsymbol{x}_{\text{state}}) \sim G(z)$ (containing a natural-language statement and corresponding hidden structured state), and the *validator* $V(\boldsymbol{x}, a) \in \{0, 1\}$ deterministically judges the correctness of a candidate answer $a$.

**Single-solution verifiability.** We focus on tasks where the hidden state $\boldsymbol{x}_{\text{state}}$ fully specifies a unique correct answer $y^*(\boldsymbol{x})$. We define a rule-based *canonical solver* $S$ such that $y^*(\boldsymbol{x}) = S(\boldsymbol{x}_{\text{state}})$. The verification process is then $V(\boldsymbol{x}, a) = \mathbb{I}[a \equiv S(\boldsymbol{x}_{\text{state}})]$, where $\mathbb{I}[\cdot]$ is the indicator function and $\equiv$ denotes equivalence after normalization.

**Checker.** To ensure data-level quality control, we also employ a lightweight auxiliary *checker* $K$ (e.g., for format validity or sanity checks), which is part of the filtering mechanism and distinct from the formal task definition $(G, V)$.

### 2.2. Synthesis Formalism

**Iterative refinement.** We model the synthesis of a task family as an iterative process $\boldsymbol{\tau} = (\mathcal{T}^{(1)}, \ldots, \mathcal{T}^{(t_{\max})})$, where $\mathcal{T}^{(t)}$ is the candidate at round $t$. The process terminates at round $t_{\max}$ when the family is accepted at finalization.

**Composite gating.** We abstract the multi-stage validation as a composite function $\Phi(\mathcal{T}) = g_1(\mathcal{T}) \wedge g_2(\mathcal{T}) \in \{0, 1\}$, where $g_1$ and $g_2$ represent quality assurance and dynamic verification, respectively, both independently carried out by separate agents. Acceptance is defined as $\Phi(\mathcal{T}^{(t_{\max})}) = 1$.

## 3. SCALING THE SCALING LOGIC

We propose SCALING THE SCALING LOGIC, an autonomous meta-algorithm designed for *meta-level scaling* of data production. Unlike traditional methods that expand static datasets, SCALING THE SCALING LOGIC synthesizes high-quality Task Families $\mathcal{T} = (G, V)$ through a rigorous,

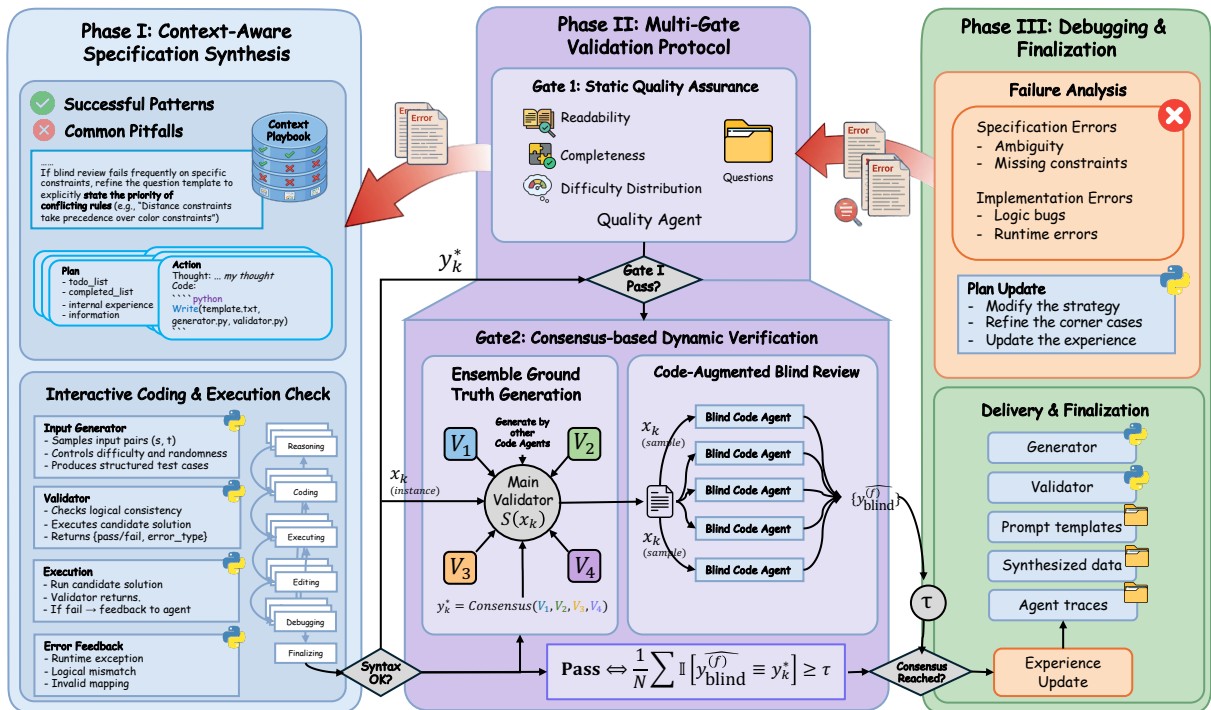

*Figure 2.* Overview of the Multi-Gate Agentic Meta-Synthesis Framework. The Main Agent operates in a three-phase closed loop: Task Synthesis (Phase I), screening via Quality Agent Gates and Consensus-based Validation (including Blind Review) (Phase II), and Abductive Debugging for failures with Experience Updates, finally delivering Generators/Validators, templates, and data (Phase III).

closed *Generate-Validate-Refine* loop.

## 3.1. Agent Framework

We build upon the open-source *Cognitive Kernel-Pro* architecture (Fang et al., 2025) as the backbone for agent execution. This framework employs a two-tier hierarchy: a **Main Agent** orchestrates high-level planning and information aggregation, while dynamically dispatching specialized **Sub-Agents** as tool nodes. Operating within a *Plan-Act-Observe* loop using executable Python code, the agents maintain a structured context (e.g., *todo_list*, *experience*) to support long-horizon reasoning. As illustrated in Figure 2, the Main Agent functions akin to a software engineer: implementing code, executing smoke tests in a terminal, and iteratively debugging based on interpreter traces. Full prompt templates for CKPro and SSLogic are in Appendix O.

## 3.2. Phase I: Context-Aware Specification Synthesis

The system initiates when the Main Agent receives a seed.

**Automated Experience Injection.** A Context Playbook containing design patterns and common pitfalls from prior runs is injected into the context of the agent as a cross-run experience. Leveraging these priors, the agent formulates a development plan, outlining the algorithmic structures for both the Generator ($G$) and Validator ($V$). We include representative experience entries in Appendix P.

**Interactive Coding and Execution Check.** The agent implements *generator.py* and *validator.py* via an interactive loop. Crucially, the agent performs *preliminary execution checks* (smoke testing) to autonomously patch syntax errors and runtime exceptions. This ensures code is syntactically robust before submitting it to logical validation.

## 3.3. Phase II: Multi-Gate Validation Protocol

To eliminate ill-formed, ambiguous, or unsolvable tasks, we instantiate the composite gating $\Phi(\mathcal{T}) = g_1(\mathcal{T}) \wedge g_2(\mathcal{T})$ defined in § 2.2, where $g_1$ targets data-level quality and $g_2$ targets dynamic verifiability. This protocol serves as a practical data-quality mechanism that filters the majority of invalid families before they can enter the training set.

**Gate 1 ($g_1$): Static Quality Assurance.** We first apply a lightweight checker $K$ to filter format violations and sanity issues, followed by a Quality Agent that reviews $\boldsymbol{x}_{\text{text}}$ for readability, completeness, and difficulty. Failure triggers immediate refinement by the Main Agent.

**Gate 2 ($g_2$): Consensus-based Dynamic Verification.** For tasks in § 2.1, the state $\boldsymbol{x}_{\text{state}}$ fully specifies a unique correct answer $y^*(\boldsymbol{x})$ via the canonical solver $S$, i.e., $y^*(\boldsymbol{x}) = S(\boldsymbol{x}_{\text{state}})$. Here, $S$ is the *theoretical* ground-truth definition. In practice, since any single synthesized solver may contain implementation bugs, we adopt **consensus across independently synthesized validators** as the *operational*

ground truth. Specifically, we instantiate a pool of $m$ independently synthesized canonical solvers $\{S^{(i)}\}_{i=1}^{m}$ and derive the ground-truth label for instance $x_k$ via majority-vote consensus:

$$y_k^* = \text{Consensus}\Big(S^{(1)}(x_{k,\text{state}}), \ldots, S^{(m)}(x_{k,\text{state}})\Big). \tag{1}$$

If no strict majority exists, the instance is deemed ambiguous, triggering a reversion to Phase I for refinement.

**Code-Augmented Blind Review.** Subsequently, we submit only the textual component $x_{k,\text{text}}$ (withholding the hidden structured state $x_{k,\text{state}}$) to $n$ independent reviewer agents, each equipped with a Python execution tool to solve the task from scratch. The instance passes blind review if reviewer solutions match $y_k^*$ at a rate at least $\tau$:

$$\text{Pass} \iff \frac{1}{n}\sum_{i=1}^{n}\mathbb{I}\Big[\hat{y}_{\text{blind}}^{(i)} \equiv y_k^*\Big] \geq \tau, \tag{2}$$

where $\mathbb{I}[\cdot]$ is the indicator function and $\equiv$ denotes equivalence after normalization (as in § 2.1).

### 3.4. Phase III: Feedback-Driven Finalization

Validation failure is treated as a structured observation. If Gate 2 fails, the Main Agent analyzes execution traces (e.g., conflict reports, reviewer logs) to distinguish ambiguity from logic bugs, updates the code, and retries. Upon passing all gates, the artifacts are packaged as canonical generators for large-scale data production.

## 4. Impact of SSLOGIC on RL

This section is organized around two core questions: (1) whether our synthesis-evolution pipeline can produce logic task streams of **higher training utility**; and (2) how these gains correspond to the **training dynamics** (e.g., the evolution of reasoning trajectory length and self-correction signals) during the optimization process.

Unless otherwise stated, all performance evaluations and statistics are conducted on the evaluation set at Table 2, using consistent decoding settings to ensure comparability. Benchmark descriptions and sizes are summarized in Appendix C (Table 11).

### 4.1. Setup

We employ a dual-model strategy using **Qwen3-8B-Base** for primary attribution analysis and **Qwen3-8B(Thinking)** for performance ceiling exploration. Training adheres to a fixed optimization step protocol using GRPO without SFT, avoiding strong model distillation, reporting results at 160, 200, and 240 steps to ensure fair comparison. Note that our Seed families are authored in Chinese while all

evaluations use English benchmarks; the size-controlled Enigmata experiment (§4.5) further confirms the pipeline is equally effective on English-only data. Dataset sizes and mixing ratios for all key settings are summarized in Appendix D. Hyperparameters and decoding configurations are listed in Appendix H. See Appendix G for detailed experimental protocols. Qualitative examples of synthesized tasks are provided in Appendix F.

### 4.2. Synthetic Data Consistency

To ensure gains from SSLOGIC-Evolve reflect training-signal quality rather than **Difficulty Drift**, we evaluated solvability (pass@1) on a paired validation set using an independent model (**Doubao-1.6-Thinking**). Results in Table 1 show that Evolve and Seed maintain **basic consistency** with no systematic difficulty collapse or surge. We report pass@1 as the primary evaluation metric; definitions and benchmark-specific sampling budgets are in Appendix B (Eq. 3, Table 10).

*Table 1.* **Assessment of Difficulty Drift.** We compare the solvability of Seed and Evolved datasets under fixed difficulty-level sampling {5,7,10} with a 1:1:1 ratio. Gray entries highlight performance inflation due to model-family alignment (DeepSeek-Chat generated tasks evaluated by DeepSeek-Reasoner). Note that despite using different variants (Chat vs. Reasoner), the shared pre-training distribution introduces bias.

| Evolver | Eval Models | Dataset | pass@1 | pass@5 | pass@8 |
|---|---|---|---|---|---|
| **o4-mini** | **DS-Reasoner** | Seed | 0.617 | 0.746 | 0.754 |
| | | Evolve | 0.610 | 0.746 | 0.781 |
| | **Doubao** | Seed | 0.605 | 0.763 | 0.798 |
| | | Evolve | 0.588 | 0.798 | 0.825 |
| **DS-Chat** | **DS-Reasoner** | Seed | 0.591 | 0.687 | 0.706 |
| | | Evolve | 0.657 | 0.755 | 0.785 |
| | **Doubao** | Seed | 0.631 | 0.767 | 0.804 |
| | | Evolve | 0.657 | 0.761 | 0.779 |

**Bias Mitigation.** To further mitigate the **homophily bias** (Table 1) which can inflate evaluation scores by $\sim$6.6% under same-family settings, we strictly decoupled models: using DeepSeek for evolution and Qwen for training.

### 4.3. Main Results

Having checked difficulty consistency, we examine whether the evolved task family (SSLOGIC-Evolve) possesses higher training value compared to Seed tasks. Table 2 summarizes the comparison under identical optimization steps.

**Result Analysis.** As shown in Table 2 (A), under fixed optimization step constraints, models trained with Evolve exhibit a more consistent upward trend on logic benchmarks (SynLogic +0.6–2.0, BBH +1.3–3.4). Simultaneously, we observe spillover gains on mathematical benchmarks (AIME24 +1.5–3.5, Math500 +1.2–3.7). This em-

*Table 2.* **Main Results on Logic and Mathematical Benchmarks under Fixed Optimization Steps.** Results are shown in four settings: (A) human-annotated Seed vs. agent-evolved tasks; (B) synergistic mathematical training with the data-mixing ladder; (C) size-controlled experiment on external Enigmata data; (D) Qwen3-8B(Thinking) training for performance-ceiling exploration. Qwen3-8B-Base is listed at the bottom. $\Delta$ denotes the average absolute improvement over the baseline row within each block, averaged across benchmarks. Results are reported as pass@1 with specific sampling budgets (Appendix B); each benchmark is evaluated over multiple instances, with counts in Appendix C (Table 11).

| Data@Step | Logic Benchmarks | | | | | | Mathematical Benchmarks | | | | | |
|---|---|---|---|---|---|---|---|---|---|---|---|---|
| | SynLogic | ARC-AGI | BBH | BBEH | Enigmata | $\Delta$ | AIME24 | AIME25 | Brumo25 | HMMT25 | Math500 | $\Delta$ |
| *(A) Seed vs. Evolve under Fixed Training Steps* | | | | | | | | | | | | |
| Seed@160 | 14.6 | 3.6 | 72.9 | 13.5 | **13.4** | – | 13.2 | 11.2 | 21.4 | 2.6 | 77.9 | – |
| Evolve@160 | **14.8** | **3.8** | **74.2** | **14.7** | **13.4** | 0.6 | **15.7** | **12.9** | **23.5** | **3.6** | **79.1** | 1.7 |
| Seed@200 | 12.9 | 1.7 | 73.5 | 13.3 | **13.5** | – | 14.1 | 11.3 | 21.5 | 2.8 | 77.0 | – |
| Evolve@200 | **16.1** | **5.0** | **74.8** | **14.8** | 12.9 | 1.7 | **16.9** | **13.2** | **23.0** | **3.6** | **79.3** | 1.9 |
| Seed@240 | 13.1 | **3.1** | 72.1 | 13.6 | **13.9** | – | 13.8 | 11.0 | 21.9 | 3.2 | 77.1 | – |
| Evolve@240 | **18.7** | **3.1** | **75.5** | **15.0** | 13.3 | 2.0 | **17.3** | **14.0** | **25.6** | **4.4** | **80.8** | 3.0 |
| *(B) Synergistic Mathematical Training with Data-Mixing Ladder* | | | | | | | | | | | | |
| AIME@160 | 16.7 | 3.8 | 75.2 | 14.8 | 13.7 | – | 24.1 | 16.1 | 29.2 | 7.7 | 83.8 | – |
| +SSLogic-Evolve@160 | **18.5** | 1.9 | **77.8** | **16.0** | **15.6** | 1.1 | **25.2** | **19.2** | **34.2** | **10.8** | **86.6** | 3.0 |
| +ARC-AGI+Evolve@160 | 16.7 | **9.3** | 74.6 | 14.4 | 14.1 | 1.0 | 15.9 | 14.9 | 26.7 | 4.0 | 80.1 | −3.9 |
| AIME@200 | 16.5 | 5.3 | 76.1 | 14.6 | 14.1 | – | 25.6 | 15.6 | 30.8 | 7.9 | 83.4 | – |
| +SSLogic-Evolve@200 | **19.5** | 2.9 | **78.9** | **16.1** | **16.3** | 1.4 | **27.0** | **21.0** | **36.0** | **12.4** | **85.5** | 3.7 |
| +ARC-AGI+Evolve@200 | **19.5** | **8.6** | 75.9 | 15.1 | 14.6 | 1.4 | 16.5 | 14.6 | 24.2 | 3.1 | 79.9 | −5.0 |
| AIME@240 | 13.3 | 3.8 | 76.5 | 15.4 | 14.4 | – | 26.5 | 15.2 | 30.4 | 7.2 | 83.5 | – |
| +SSLogic-Evolve@240 | **22.1** | 1.9 | **79.5** | **16.6** | **18.5** | 3.0 | **29.7** | **22.4** | **37.0** | **13.0** | **87.8** | 5.4 |
| +ARC-AGI+Evolve@240 | 20.0 | **11.2** | 76.1 | 15.6 | 15.4 | 3.0 | 18.4 | 14.7 | 26.4 | 3.6 | 81.2 | −3.7 |
| *(C) Size-Controlled Experiment on External Data (Enigmata)* | | | | | | | | | | | | |
| Enigmata-Seed@240 | 22.3 | 1.7 | 61.8 | 16.3 | – | – | 21.3 | 11.5 | 20.2 | 4.2 | 83.6 | – |
| Enigmata-Evolve@240 | **24.3** | **1.9** | **67.3** | **17.2** | – | 2.4 | **23.4** | **16.4** | **21.2** | **4.4** | 82.7 | 1.6 |
| *(D) Training on Qwen3-8B(Thinking)* | | | | | | | | | | | | |
| Seed@240 | 49.4 | 3.8 | 89.2 | 30.1 | 38.9 | – | **74.3** | 62.9 | **71.4** | **42.8** | 96.2 | – |
| Evolve@240 | **52.1** | **7.9** | **89.3** | **31.1** | **39.5** | 1.7 | 74.1 | **65.5** | 70.9 | 42.0 | **96.3** | 0.2 |
| Qwen3-8B(Thinking) | 51.1 | 7.6 | 89.3 | 30.0 | 39.4 | – | 71.7 | 63.2 | 68.7 | 40.9 | 96.1 | – |
| Qwen3-8B-Base | 8.4 | 1.4 | 60.9 | 10.1 | 7.5 | – | 6.8 | 5.6 | 11.8 | 0.8 | 60.4 | – |

pirical pattern is consistent with the automated evolution pipeline generating a collection of instances with higher effective training value under fixed steps.

**Converging Evidence.** Three complementary comparisons support this conclusion: (1) the step-matched results above (Table 2); (2) a token-budget analysis showing that, at matched cumulative RL tokens ($\sim$1.09B), Evolve still outperforms Seed by +7.3% relative Macro Accuracy (details in Appendix N); and (3) a size-controlled experiment on the external Enigmata benchmark (§4.5). Together, these three converging comparisons suggest higher training utility of the evolved task families, ruling out data scale, compute budget, and seed distribution as primary confounds.

### 4.4. Isolating the Effect of Dataset Size

To isolate the impact of task quality from potential confounding factors such as dataset size, we conducted a strictly paired experiment using a subsampled dataset. We selected 236 human-annotated Seed problems and 236 corresponding Evolve variants (all with $d = 7$). As shown in Table 3, models trained on Evolve data consistently outperform those

trained on Seed data even under identical sample counts, with an average logic gain of +0.8 and a math gain of +1.2. This confirms that the observed improvements in §4.3 are primarily driven by the higher effective training value of the evolved task structures rather than mere increases in data quantity. Full benchmark details for this subsampled experiment are provided in Appendix I.

*Table 3.* **Delta Comparison on Subsampled Data** ($n = 236$ **versus** 236**).** The table shows the average improvement ($\Delta$) of Evolve over Seed at different training steps.

| Metric | Step 160 | Step 200 | Step 240 |
|---|---|---|---|
| Logic $\Delta$ | +1.0 | +1.2 | +0.3 |
| Math $\Delta$ | +0.7 | +1.8 | +1.0 |

### 4.5. Size-Controlled Experiment on External Data

To test generalization beyond our Seed distribution, we applied the Evolve pipeline to Enigmata (Chen et al., 2025a), an independent English-only logic puzzle benchmark. We selected 30 of 36 types (those with parameterized difficulty) and evolved them, matching instance counts (8,776 vs 8,758)

and training steps (240). With all controls matched, Evolve yields consistent gains: BBH +5.5, AIME25 +4.9, SynLogic +2.0, BBEH +0.9 (Table 2 C). This confirms the pipeline is seed-agnostic and requires zero human annotation. Full protocol details are in Appendix M.

### 4.6. Cross-Domain Generalization

This section investigates the synergistic impact of incorporating synthetic logic data into high-resource mathematical training. Specifically, we integrate SSLOGIC into the mathematics training set (consisting of AIME 1983–2023 problems, excluding image-based ones), following the experimental setup of Enigmata (Chen et al., 2025a).

As shown in Table 2 (B), we observe a divergent trend between task types. After adding SSLOGIC-Evolve data, the model achieves better OOD performance on tasks such as AIME24/25 and Math500 (+3.0–3.7 average gain). In contrast, mixing **ARC-AGI** data into the Math + Evolve set is associated with performance drops on math benchmarks (e.g., -3.9 to -5.0 on AIME24/25).

This outcome suggests that different reasoning tasks vary in their utility for robust general logic transfer. The abstract reasoning patterns in SSLOGIC (e.g., constraint satisfaction via explicit steps) appear to serve as effective reasoning scaffolding. Conversely, the implicit pattern matching in ARC may instead interfere with the rigorous step-by-step derivation required for mathematics.

### 4.7. Training Dynamics

We analyze the changes in the reasoning trajectories of the model during training, focusing on the co-evolution of **response length** and **self-checking language signals**.

Following DeepSeek-R1 (DeepSeek-AI et al., 2025a), we monitor the frequency of specific reflection-like tokens (e.g., *wait*, *mistake*, *check*) on the validation set to track the emergence of self-correction behaviors. See Appendix J for the full vocabulary and statistical details.

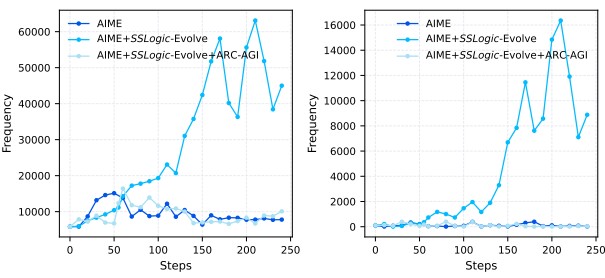

*Figure 3.* Evolution of reflection-like token frequency across different training settings.

As shown in Figure 3, after introducing logic training, the model exhibits **staged length growth**, accompanied by a

parallel rise in reflection-like token frequency. This suggests that SSLOGIC training systematically reshapes reasoning traces beyond final metrics. When controlling for response length, the gap persists: Evolve produces 27.80 reflection tokens per 100 response tokens versus Seed's 17.57 (+58%), confirming that the increased reflection is not merely an artifact of longer outputs. In contrast, the ARC-AGI-mixed setting (Math+ARC-AGI+Evolve) shows weaker length growth and a flatter reflection-like token curve (see § 4.8). This divergence aligns with the cross-domain performance gap, indicating that longer trajectories correlate positively with cross-domain performance overall.

### 4.8. Negative Control

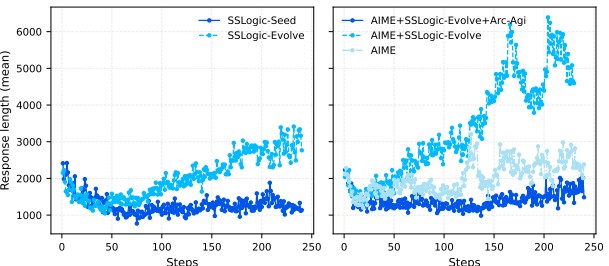

*Figure 4.* Average response length dynamics during training.

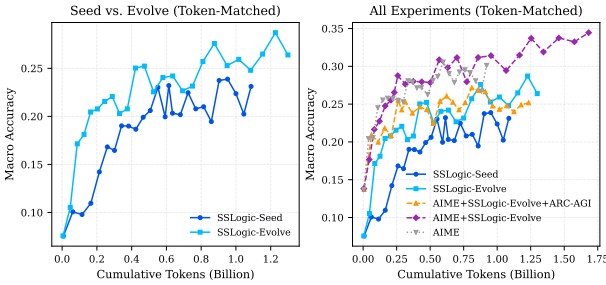

*Figure 5.* Macro Accuracy vs. cumulative RL tokens. At matched token budgets (∼1.09B), Evolve maintains a +7.3% relative advantage over Seed, confirming that the gains are not solely attributable to additional compute from longer responses.

To understand the impact of different task structures on reinforcement learning dynamics, we analyze ARC-AGI as a **negative control** in our study.

- **Observation:** As shown in Figure 4 (Right), contrary to the length growth in the SSLOGIC group, the average response length of Math + ARC + Evolve is noticeably suppressed during training; correspondingly, its reflection-like token curve is flatter (Figure 3).

- **Analysis:** ARC tasks prioritize input-output mappings and lack explicit intermediate reasoning chains in the supervision. This structure **incentivizes** the RL policy to bypass step-by-step derivation in favor of short-path pattern matching. This contrasts with mathematical reasoning, explaining why ARC-mixed models fail to generalize to tasks requiring rigorous, multi-step deduction. Recent work further argues that ARC-AGI is fundamentally a

vision task (Hu et al., 2025), suggesting that the negative transfer stems from a domain mismatch (visual pattern recognition vs. symbolic reasoning) rather than a generic format effect.

# 5. Analysis

We dissect SSLOGIC to uncover the drivers of its training effectiveness. Comparing the *Evolved* generator–validator programs against the *Seed* baseline using AST-based metrics, we validate difficulty controllability (§ 5.1), quantify the expansion in algorithmic diversity (§ 5.2), and attribute the downstream gains to these structural shifts (§ 5.3). Finally, we examine the efficiency and cost-effectiveness of the synthesis pipeline (§ 5.4).

## 5.1. Difficulty Controllability and Solvability

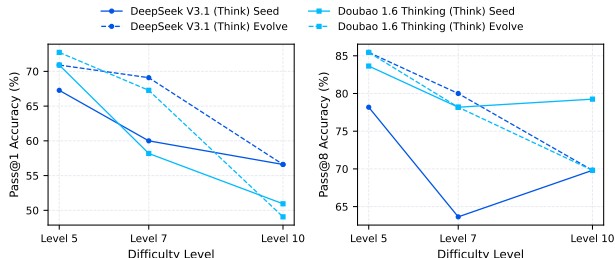

*Figure 6.* Difficulty controllability. Pass@1 accuracy between Seed and Evolved tasks at $d \in \{5, 7, 10\}$ on DeepSeek-V3.1-Terminus and Doubao-1.6-Thinking. The curves decrease monotonically and closely track each other, with error bars shown.

A key risk in synthetic generation is difficulty collapse (models default to trivial patterns) or complexity explosion (tasks become unsolvable). A controllable generator should allow smooth adjustment of difficulty for curriculum learning.

To verify this, we evaluated pass@1 of manual *Seed* tasks and agent-synthesized *Evolved* tasks across three difficulty levels ($d \in \{5, 7, 10\}$), which scale core logical constraints (e.g., graph size, recursion depth, or constraint density). We use **DeepSeek-V3.1-Terminus** and **Doubao-1.6-Thinking** as reference solvers, and report the exact pass@1 and pass@8 values by difficulty in Appendix J.

Figure 6 shows a monotonic drop in pass@1 as difficulty increases, indicating that the synthesis recipe respects the intended complexity parameters. The *Evolved* curves track the *Seed* baseline with small bidirectional fluctuations (Table 1), with no systematic difficulty collapse or explosion.

## 5.2. Algorithmic Diversity and Coverage

To quantify algorithmic diversity, we compute structural and computational metrics on Seed and paired generator–validator code from the Evolved pipelines. Figure 7 summarizes distributions of LOC, cyclomatic and cognitive com-

plexity, Halstead effort, and loop structure (computed on successfully parsed code). Detailed definitions of these metrics and additional analysis are provided in Appendix K.

Relative to Seed, all three Evolved pipelines show higher control-flow complexity (cyclomatic and cognitive) and greater loop activity; DeepSeek-chat and GLM-4.6 are also longer and higher in Halstead effort, while o4-mini remains closer in length but still increases control-flow complexity.

The inferred time-complexity distribution shifts accordingly (Figure 8, left). Seed retains a larger cubic share, while the Evolved pipelines tilt toward quadratic and graph-style regimes, especially GLM-4.6.

Pattern detectors show broader coverage of core primitives (Figure 8, right). Sorting, DP, graph traversal, and recursion appear more frequently in the Evolved pipelines, while binary search remains high across all sources and divide-and-conquer remains rare. This suggests the Evolved pipeline expands algorithmic diversity beyond ubiquitous primitives, rather than merely amplifying existing biases. We note that these AST-based metrics capture syntactic patterns and may not fully reflect semantic novelty; code-embedding-based similarity measures were explored but proved unreliable across heterogeneous code styles. The category-selective downstream gains in §5.3 provide complementary evidence that these structural differences translate into meaningfully different training signal.

## 5.3. Linking Structure to Downstream Gains

The structural shifts documented above—higher cognitive complexity, broader algorithmic patterns, and a tilt toward quadratic/graph regimes—are *associated with* selective downstream improvements rather than uniform gains.

**KORBench Category-Selective Gains.** We evaluated both models on KORBench (Ma et al., 2025), covering five reasoning categories. Gains concentrate in Logic (+13.2%) and Operation (+9.6%)—categories exercising DP, graph traversal, and complex loops—while Puzzle (+1.2%), testing spatial reasoning outside the evolution domain, barely moves (Table 4).

**BBH Sub-Task Breakdown.** The same selective pattern appears within BBH (overall: 59.4→69.3, +9.9). The top gains concentrate in multi-step tracking: penguins_in_a_table +56.2, colored_objects +48.4, shuffled_objects (7) +44.0. Sub-tasks near ceiling remain stable (boolean_expressions 93.6→94.4). A generic data-quality effect would produce uniform gains; the observed selectivity supports the hypothesis that expanded structural complexity provides a richer training signal for multi-step reasoning.

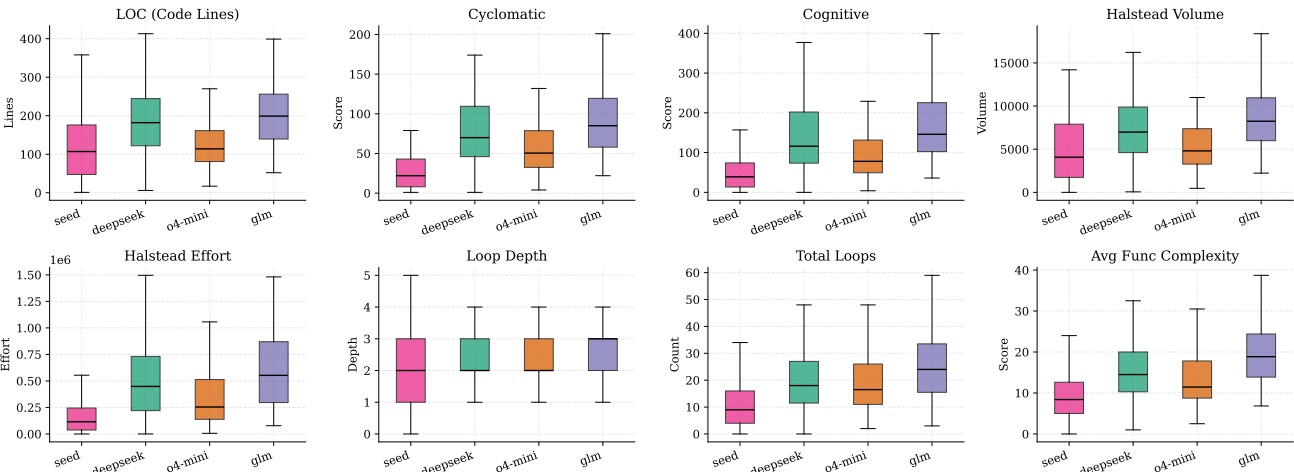

*Figure 7.* Code-level complexity across sources. Distributions of structural and computational metrics on Seed and paired generator–validator programs.

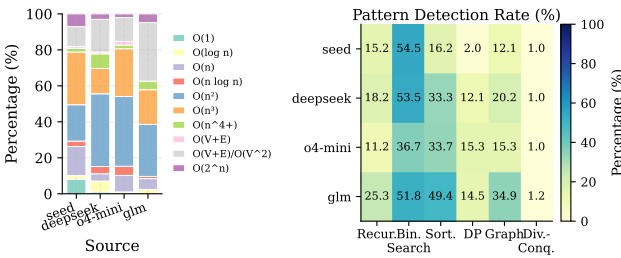

*Figure 8.* Left: Inferred time-complexity distribution. Seed shows more cubic mass, while Evolved pipelines shift toward quadratic regimes. Right: Algorithmic pattern coverage. Evolved pipelines show higher rates of sorting, DP, graph, and recursion patterns.

*Table 4.* **KORBench per-category results** (step 240, pass@1).

| Category | Seed | Evolve | Δ |
|---|---|---|---|
| Logic | 32.4 | 45.6 | **+13.2** |
| Operation | 80.8 | 90.4 | **+9.6** |
| Cipher | 20.0 | 29.2 | +9.2 |
| Counterfactual | 74.4 | 81.2 | +6.8 |
| Puzzle | 7.2 | 8.4 | +1.2 |
| **Overall** | 43.0 | 51.0 | **+8.0** |

### 5.4. Pipeline Analysis and Efficiency

Finally, we analyze the efficiency of the SCALING THE SCALING LOGIC pipeline using 100 synthesis traces produced by DeepSeek-V3.1-Terminus. The pipeline achieves a high overall acceptance rate of **55.0%** (55/100). As shown in Table 5, the two-stage gating mechanism $\Phi$ is essential for distinguishing valid logic from degenerate behaviors. Granular gate pass rates (Table 6), step-depth distributions (Table 7), runtime breakdowns (Table 8), and pass-count distributions (Table 9) are provided in Appendix A.

**Resource Divergence.** A key finding is the significant divergence in resource consumption based on the outcome of $\Phi$. Traces that fail to satisfy $\Phi$ (rejections) are disproportionately expensive, requiring **6.5× the runtime** of accepted

traces (10,265s versus 1,571s, Panel B). This confirms that invalid or ill-posed task families often trigger long-running loops or complex failure modes in solvers. By enforcing early-exit gating through $g_1$ and $g_2$, SCALING THE SCALING LOGIC effectively prevents these expensive failures from polluting the training set.

**Cost and Yield.** The amortized cost per accepted task family (where $\Phi(\mathcal{T}) = 1$) is **\$1.18**. This includes the costs incurred during the $t_{max}$ refinement rounds and the filtering of invalid candidates. Notably, 88% of the total budget is consumed by the gating agents (o4-mini) rather than the generation phase, highlighting that the primary cost driver is the multi-strategy verification (consensus and blind review) rather than code synthesis itself.

**Failure Attribution.** Analysis of traces where $\Phi = 0$ (Panel C) reveals that **Implementation Bugs** (50%) and **Unsolvability** (43%) are the main barrier. The dual-gate structure ensures that tasks passing both static quality checks and dynamic verification meet the required reasoning standards, effectively eliminating logically flawed families.

## 6. Related Work

**Logic Reasoning Benchmarks and Training.** Systematic evaluation of logical reasoning has driven diverse benchmarks, including graph and algorithmic reasoning testbeds (Parmar et al., 2024; Gui et al., 2025; Liu et al., 2021; 2023; Helwe et al., 2022; Luo et al., 2024; Suzgun et al., 2023; Kazemi et al., 2025; Ma et al., 2025; liu et al., 2025; Tang et al., 2025; Chen et al., 2024a; Zhang et al., 2024). On training, Logic-RL (Xie et al., 2025) shows RL on verifiable logic tasks unlocks reasoning capabilities, while graph/code process rewards, execution supervision, and cross-domain RL studies show transfer benefits (Peng et al., 2025; Bao et al., 2025; Chen et al., 2026; Li et al.,

*Table 5.* **Pipeline Diagnostics.** Analysis of 100 sampled tasks showing gate pass rates (Panel A), resource consumption gap (Panel B), and failure attribution (Panel C). *Rejected* traces consume significantly more compute (Runtime/Steps), highlighting the efficiency of early gating.

*Panel A: Gate Pass Rates (N=100)*

| Stage | Count | Rate | Notes |
|---|---|---|---|
| Input samples | 100 | – | – |
| Gate 1: Static QA pass | 67 | 67.0% | agentic quality check |
| Gate 2: Consensus OK | 93 | 93.0% | validator consensus |
| - Blind review pass | 55 | 55.0% | accepted $\tau \geq 3/5$ |
| Overall acceptance | **55** | **55.0%** | – |

*Panel B: Resource Divergence (Acc. vs. Rej.)*

| Metric | Accepted | Rejected | Ratio |
|---|---|---|---|
| Runtime (Avg) | 1,571s | 10,265s | **6.5×** |
| Runtime (P99) | 5,161s | 47,286s | 9.2× |
| Main Steps (Avg) | 9.7 | 16.2 | 1.7× |
| Gate Steps (Avg) | 19.8 | 34.2 | 1.7× |
| All Steps (Avg) | 29.5 | 50.3 | 1.7× |
| Total Cost (Avg/Trace) | $0.49 | $0.86 | 1.8× |

*Panel C: Failure Modes (N=44 observed)*

| Category | Count | Share | Top Issues |
|---|---|---|---|
| Implementation Bug | 22 | 50.0% | *Validator/Syntax* |
| Unsolvable / Other | 19 | 43.2% | *Logic Gap* |
| Ambiguity | 3 | 6.8% | *Vague Spec* |

2025b). To the best of our knowledge, leveraging scalable task-family synthesis, we provide one of the first reports on synthesizing and training with a near-thousand-scale collection of verifiable logic task families, together with a controlled analysis of code-level evolution, training dynamics, and cross-benchmark generalization.

**Synthetic Data for Logic Reasoning.** Prior synthesis generally falls into two paradigms: (1) *formal verifiers*, such as SLR (Helff et al., 2025) targeting inductive logic programming and ProtoReasoning (He et al., 2025) using prototype-based generation; and (2) *human-authored templates*, where works like Reasoning Gym (Stojanovski et al., 2025), Enigmata (Chen et al., 2025a), SynLogic (Liu et al., 2025), Math-Octopus (Chen et al., 2024c), and ControlMath (Chen et al., 2024b) scale task or math-data production but rely on fixed sources, templates, or generators. AutoLogi (Yu et al., 2025) integrates LLMs for rewriting but focuses on SAT problems. Unlike these methods often constrained to specific domains (e.g., ILP, SAT) or fixed templates, our framework leverages code agents to enable scalable synthesis for general algorithmic reasoning.

**Agentic Synthesis Pipelines.** AgentFrontier (Chen et al., 2025b) and AgentEvolver (Zhai et al., 2025) enable autonomous task discovery and evolution through feedback. Similar pipelines like AgenticMath (Liu et al., 2026), Agentic Proposing (Jiao et al., 2026b), Socratic-Zero (Jiao et al., 2026a), SPICE (Liu et al., 2025a), and InfoSeek (Xia et al.,

2025) target mathematical and research tasks, while Zhou et al. (2025) focus on competitive programming. Following this trend, we propose *meta-synthesis* for logic tasks, where the key difficulty lies in *reliable verification*: logical correctness is often hard to check automatically and is naturally decoupled from surface-form reasoning traces, motivating multi-gate validation and adversarial review.

## 7. Limitations

While our results demonstrate the effectiveness of agentic meta-synthesis for scaling verifiable training signals, several limitations warrant discussion.

Our experiments are scoped to logic and mathematical reasoning; whether the framework yields comparable gains in other verifiable domains such as code generation or scientific reasoning remains to be validated. Similarly, all RL training is conducted on Qwen3-8B variants—the framework is model-agnostic by design, but cross-family verification (e.g., LLaMA, Mistral) is left for future work.

The evolution pipeline requires a seed set with generator–validator structure to bootstrap. The Enigmata experiment (§4.5) demonstrates seed-agnostic improvements on external data, but cold-start synthesis without any structured seed is not yet supported.

On the evaluation side, while Section 5.4 quantifies each gate's filtering load, a full ablation that trains on data filtered by each gate independently would require multiple expensive RL runs and has not yet been conducted. Additionally, our step-matched comparison does not fully control for compute: Evolve elicits longer responses, consuming ∼1.20× the tokens at 240 steps. The token-matched analysis (Appendix N) shows that roughly half the step-matched gain persists after equalizing token budgets, but a fully compute-matched comparison is not performed.

Finally, the pipeline's 55% acceptance rate is bounded by the synthesis model's code-generation capability. Rejected families tend to be harder, introducing a coverage ceiling and survivorship bias toward families within the current generation frontier.

## 8. Conclusion

We propose SCALING THE SCALING LOGIC, an agentic meta-synthesis framework for scaling verifiable training signals for RLVR. Three converging comparisons—step-matched, token-matched, and size-controlled on external data—consistently indicate higher training utility of evolved specifications, driving gains in both logical and cross-domain mathematical reasoning. We open-source the complete framework and datasets to support reproducible research.

## Acknowledgements

This work is supported by NSFC 62572418 and Guangdong Provincial Talent Program (No. 2024TQ08X366). The first author thanks Yuhan Li, Miao Peng, and Nuo Chen for their thoughtful critiques of the ideas in this work, their constructive pushes and guidance on future research directions, and their prior work and mentorship that helped shape the first author's path toward LLM research.

## Impact Statement

This paper introduces SSLogic, an agentic framework for autonomously synthesizing verifiable reasoning tasks to scale the training of logic-driven large language models. The primary societal impact of this work lies in its potential to advance the reliability and reasoning capabilities of AI systems without relying on labor-intensive human annotation. By providing a scalable source of high-quality, verifiable training signals, our method contributes to the democratization of advanced AI development and promotes the creation of models that are grounded in rigorous logic rather than superficial pattern matching.

However, as with any technology that enhances general reasoning capabilities, there are potential risks. The improved planning and logic skills could theoretically be misused to automate complex malicious tasks. Additionally, while our agentic synthesis reduces direct human bias, the reliance on seed families and base models for evolution means that subtle biases could still be propagated or amplified. We mitigate these risks through our multi-gate verification protocol, which strictly enforces correctness and solvability, reducing the likelihood of hallucinations and unreliable outputs. We advocate for responsible deployment and continuous monitoring of synthesized datasets to ensure they remain diverse, unbiased, and safe for downstream applications.

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

# A. Detailed Pipeline Analysis Statistics

In Section 5.4, we provided a summary of the efficiency of the generation pipeline. Here, we present the granular data regarding gate pass rates, step depth distributions, and runtime statistics for the analyzed batch of 100 tasks.

## A.1. Gate Pass Rates

Table 6 details the specific pass rates for each quality gate. QQC (Question Quality Check) serves as the initial filter for format and completeness, followed by a consensus check and a final blind review.

*Table 6.* Detailed gate pass rates for the pipeline.

| Gate Stage | Count | Pass Rate |
|---|---|---|
| Gate 1 (QQC pass) | 67 / 100 | 67.0% |
| Gate 2 (Consensus, validator OK) | 93 / 100 | 93.0% |
| Gate 2 (Blind review pass) | 55 / 100 | 55.0% |
| Overall acceptance | 55 / 100 | 55.0% |

## A.2. Step Depth Analysis

We analyze the reasoning depth by counting `CHAIN` spans. Table 7 separates steps executed by the main agent (Seed-Main) versus the quality gating agents (Seed-Quality/Blind). Rejected tasks typically exhibit higher variance and maximum step counts, indicating failure loops.

*Table 7.* Distribution of step depths for accepted versus rejected traces.

| | Main Steps | | | Gate Steps | | |
|---|---|---|---|---|---|---|
| | Average | Median | 99th Percentile | Average | Median | 99th Percentile |
| Accepted | 9.67 | 9 | 22 | 19.78 | 14 | 63 |
| Rejected | 16.18 | 17.5 | 24 | 34.16 | 25 | 103 |

## A.3. Runtime Statistics

Table 8 provides the breakdown of runtime distributions. The 99th percentile runtime for rejected traces is nearly an order of magnitude higher than accepted ones.

*Table 8.* Runtime statistics (seconds) for accepted, rejected, and overall traces.

| Status | Average (s) | Median (s) | 99th Percentile (s) |
|---|---|---|---|
| Accepted | 1570.8 | 1103.1 | 5161.1 |
| Rejected | 10264.8 | 3922.5 | 47286.0 |
| Overall | 5434.8 | 2057.2 | 47282.9 |

## A.4. Blind Review Pass-Count Distribution

We summarize the number of samples passing blind review (`pass_count`) among the 70 tasks with recorded blind-review outcomes.

# B. Evaluation Metrics

To evaluate the reasoning capabilities of our models, we primarily use the **pass@$k$** metric. For a given problem, we generate $n$ independent samples and determine the number of correct samples $c$. The unbiased estimate of pass@$k$, as proposed by (Chen et al., 2021), is calculated as:

$$\text{pass@}k = \mathbb{E}\left[1 - \frac{\binom{n-c}{k}}{\binom{n}{k}}\right] = 1 - \frac{\binom{n-c}{k}}{\binom{n}{k}} \tag{3}$$

*Table 9.* Distribution of blind-review pass counts (N=70).

| Pass Count | 0 | 1 | 2 | 3 | 4 | 5 |
|---|---|---|---|---|---|---|
| Tasks | 11 | 2 | 2 | 12 | 11 | 32 |

In this paper, we report pass@1 for all evaluation benchmarks, using a benchmark-specific sampling budget $n$ (Table 10). When $k = 1$, this simplifies to the standard pass rate: pass@1 $= \frac{c}{n}$. For diagnostic analyses (e.g., difficulty-level solvability), we also report pass@8 using the same definition.

*Table 10.* Evaluation benchmarks and sampling budgets for pass@1.

| Benchmark | Samples per Problem ($n$) |
|---|---|
| MATH-500 | 8 |
| AIME 2024 | 64 |
| AIME 2025 | 64 |
| HMMT 2025 | 64 |
| BRUMO 2025 | 64 |
| ARC-AGI | 1 |
| BBH | 1 |
| BBEH | 1 |
| SynLogic Val | 1 |
| Enigmata Eval | 1 |

## C. Benchmark Details

*Table 11.* Number of problems in each evaluation benchmark.

| Benchmark | # Problems |
|---|---|
| MATH-500 | 500 |
| AIME 2024 | 30 |
| AIME 2025 | 30 |
| HMMT 2025 | 30 |
| BRUMO 2025 | 30 |
| ARC-AGI | 419 |
| BBH | 6511 |
| BBEH | 1200 |
| SynLogic Val | 1000 |
| Enigmata Eval | 4758 |

Here we introduce the benchmarks used in this work. The sizes of the test sets are reported in Table 11. Some descriptions are adapted from (Bao et al., 2025). The following are established benchmarks used by the community.

### C.1. Mathematics

**MATH-500** (Lightman et al., 2024; Hendrycks et al., 2021) A 500-problem held-out subset of the MATH benchmark, commonly used as an evaluation set for competition-level mathematical reasoning.

**AIME24 and AIME25** (Veeraboina, 2024; OpenCompass, 2025) Problem sets from the American Invitational Mathematics Examination (AIME). AIME problems are short-answer contest questions that require non-trivial mathematical reasoning and multi-step derivations.

**HMMT-2025 (February)** (Balunović et al., 2025) Problems from the February 2025 Harvard–MIT Mathematics Tournament (HMMT), a well-known high-school mathematics competition featuring challenging and creative problem solving.

**BRUMO-2025** (Balunović et al., 2025) Problems from the 2025 Brown University Mathematical Olympiad (BRUMO), evaluating advanced mathematical reasoning and solution construction.

## C.2. Logical Reasoning

**ARC-AGI** (Chollet, 2024; Chollet et al., 2025) The Abstraction and Reasoning Corpus (ARC) is a grid-based reasoning benchmark designed to measure the efficiency of acquiring new abstract skills from only a few examples. We evaluate on ARC-AGI-1.

**BBH** (Srivastava et al., 2023; Suzgun et al., 2023) BBH (Beyond the Imitation Game Benchmark) contains 23 challenging tasks selected from BIG-bench, covering diverse forms of reasoning such as multi-step logical deduction, algorithmic/procedural reasoning, and compositional generalization.

**BBEH** (Kazemi et al., 2025) BIG-Bench Extra Hard (BBEH) provides more difficult counterparts to BBH tasks by constructing novel, harder task variants, aiming to stress-test advanced reasoning capabilities.

**Enigmata-eval** (Chen et al., 2025a) A puzzle-reasoning benchmark proposed in the Enigmata suite. It features diverse puzzle tasks equipped with rule-based verifiers, enabling automatic and objective evaluation under well-defined task rules.

**Synlogic-val** (Liu et al., 2025) A synthetic logical reasoning validation set from the SynLogic framework. It covers a broad set of logical task types with rule-based verification, supporting controlled evaluation of deduction under explicit task rules.

# D. Dataset Sizes and Mixing Ratios

Table 12 summarizes the training set sizes and compositions for the key settings referenced in § 4. We report the Seed/Evolve setting and the math ablations (Math only, Math+Evolve, Math+ARC-AGI+Evolve).

For Evolve, we evolve Seed problems with DeepSeek+o4-mini using three independent samplings, keep instances with agreement threshold $\tau \geq 3/5$, then sample by difficulty buckets to match the Seed distribution, with $(0\text{–}3) : (4\text{–}6) : (7\text{–}10) \approx 1 : 3 : 5$. We upsample within buckets to reach this target ratio, yielding 15,671 instances. Seed is manually annotated and therefore smaller, so we do not enforce exact size alignment between Seed and Evolve. We do not observe significant difficulty drift between Seed and Evolve (Table 1). Both Seed and Evolve pools are filtered with a CodeI/O-style checker (following (Li et al., 2025a)) to remove instances with very short execution time (e.g., $< 0.1\,s$), which reduces trivial tasks. The math proportions in these ablations are 18.5%, 50.0%, and 100%, respectively.

*Table 12.* Training dataset sizes and mixing ratios for key settings. Counts are number of training instances; percentages in parentheses.

| Setting | Total | Seed | Evolve | AIME | ARC-AGI |
|---|---|---|---|---|---|
| Seed | 5,718 | 5,718 (100%) | – | – | – |
| Evolve | 15,671 | – | 15,671 (100%) | – | – |
| Math+ARC-AGI+Evolve | 31,342 | – | 15,671 (50.0%) | 5,796 (18.5%) | 9,875 (31.5%) |
| Math+Evolve | 31,342 | – | 15,671 (50.0%) | 15,671 (50.0%) | – |
| Math only | 31,342 | – | – | 31,342 (100%) | – |

# E. Seed Task Type Distribution

We summarize Seed task diversity manually. A small portion of entries have missing or non-standard paths and are grouped as "Unlabeled/Other".

*Table 13.* Seed task type distribution

| Task Type | Count | Share |
|---|---|---|
| Logical Reasoning | 135 | 34.0% |
| Spatial Reasoning | 85 | 21.4% |
| Symbolic Reasoning | 72 | 18.1% |
| Temporal Reasoning | 38 | 9.6% |
| Relational Reasoning | 9 | 2.3% |
| Commonsense Reasoning | 7 | 1.8% |
| Causal Reasoning | 3 | 0.8% |
| Unlabeled/Other | 48 | 12.1% |

# F. Example Tasks (Seed versus Evolve)

We present full, executable examples (Seed versus Evolved) for major task categories to demonstrate how the synthesizer expands task complexity while ensuring verifiability.

## F.1. Logical Reasoning: Bacterial Infection

---

**Seed Task: Bacterial Spread (Standard)**

**Problem Description:** A Petri dish is modeled as a grid (rows top-down from 0, cols left-right from 0). The number in each cell represents its nutrient level:

```
7 5 4
6 7 5
8 4 5
```

Bacteria are initially injected at $(1, 1)$ (quantity 3) and $(1, 0)$ (quantity 2). Bacteria infect cells over time. **Rules:** 1. Upon injection, infected count = injected quantity + nutrient level. 2. If an infected cell has $\geq 3$ bacteria, it spreads to **one** uninfected neighbor (Up, Left, Right, Down) that has the **maximum nutrient level**. Tie-break priority: Up $>$ Left $>$ Right $>$ Down. 3. The spreading cell sends $\lfloor current\_count/3 \rfloor$ bacteria to the target. The count of the target becomes: received bacteria + its own nutrient level. The count of the sender decreases by the sent amount. 4. Process continues until no further spread is possible.

**Question:** What is the final bacteria count distribution in the grid?

---

---

**Evolved Task: Bacterial Spread (Complex)**

**Problem Description:** In a grid (rows/cols from 0), values denote nutrient levels:

```
1 6 4 2 2 1
2 2 5 8 2 1
3 1 6 1 3 5
1 7 7 5 4 4
5 7 7 3 2 7
5 8 2 8 6 7
```

Initial bacteria injections: - $(5, 1)$: 5 units - $(0, 0)$: 1 unit - $(3, 0)$: 2 units - $(3, 1)$: 1 unit

**New Rules:** 1. **Initial State:** Cell count = Injected amount + Nutrient level. 2. **Spread Condition:** A cell spreads only if its bacteria count $\geq 10$. 3. **Target Selection:** Spread to the uninfected neighbor with maximum nutrient among **8 neighbors** (Up, Down, Left, Right, and 4 Diagonals). Tie-break: Up $>$ Down $>$ Left $>$ Right $>$ Top-Left $>$ Top-Right $>$ Bottom-Left $>$ Bottom-Right. 4. **Transfer Logic:** The sender sends $\lfloor current\_count/3 \rfloor$ bacteria. The count of the sender decreases by this amount. The count of the target becomes: received amount + its nutrient level. 5. **Termination:** Repeat until no new cells can be infected.

**Question:** What is the maximum bacteria count in any single cell in the final grid?

---

**Analysis.** The evolution introduces three key dimensions of complexity:

- **Topology:** The neighbor definition expands from 4-connectivity (Von Neumann) to 8-connectivity (Moore), significantly increasing the branching factor of the state space.
- **State Dynamics:** The spread threshold is raised ($3 \to 10$) and the grid size is doubled ($3 \times 3 \to 6 \times 6$), requiring longer-horizon simulation to reach equilibrium.
- **Multi-source Interaction:** Multiple injection points create competing infection fronts, testing the ability of the solver to handle concurrent state updates correctly.

## F.2. Spatial Reasoning: Pattern Inference

---

**Seed Task: Linear Movement**

**Problem Description:** Infer the pattern from the following sequence of $3 \times 3$ grids containing squares (■) and circles (●).

```
Grid 1:     Grid 2:     Grid 3:     Grid 4:
|..o|       |..o|       |...|       |...|
|...|   ->  |..o|   ->  |...|   ->  |...|
|..x|       |...|       |..o|       |o.x|
```

(Note: Simplified ASCII representation for brevity. Actual input uses standard matrix format.)
**Rules:** 1. The Square (■) moves 1 step to the right per frame, wrapping around to the start of the row if it hits the edge. 2. The Circle (●) remains static.
**Question:** Calculate the total number of shapes in the 3rd row and 3rd column of the 5th figure.

---

**Evolved Task: Boundary & Symmetry**

**Problem Description:** A sequence of $3 \times 3$ matrices connected by →: `[[1,1,1],[0,1,0],[1,1,1]] -> [[1,0,1],[1,1,1],[1,0,1]]` ... (where 1=Empty, 0=Circle, Square distinct). The shapes follow a hidden movement logic.
**New Rules:** 1. **Boundary Movement:** The Square moves **clockwise along the grid boundary** by **2 steps** per frame. (It never enters the center cell $(1, 1)$). 2. **Symmetry Constraint:** The Circle is always positioned **centrally symmetric** to the Square relative to the grid center $(1, 1)$. - Example: If Square is at $(0, 0)$, Circle must be at $(2, 2)$.
**Question:** Based on this law, how many Squares and Circles are there in total in the 3rd row and 3rd column of the 5th figure?

---

**Analysis.** The evolved task shifts from simple independent linear motion to coupled, constraint-based motion:

- **Trajectory Complexity:** The "boundary clockwise" movement is a non-linear path in Cartesian coordinates, requiring the solver to model the grid topology (edges versus interior) rather than simple $(x + 1)$ arithmetic.
- **Relational Constraint:** The position of the Circle is no longer independent but functionally dependent on the position of the Square ($\mathbf{p}_{\text{circle}} = f_{sym}(\mathbf{p}_{\text{square}})$), forcing the solver to deduce global geometric relations.

## F.3. Symbolic Reasoning: Operation Optimization

---

**Seed Task: Frequency Maximization**

**Problem Description:** In a class, students have the following sticker counts: `[56, 49, 53, 24, 60]` You have **4 operations**. In each operation, you can choose any student and increase or decrease their sticker count by 1.
**Question:** What is the maximum number of students who can end up with the **exact same** number of stickers after at most 4 operations?

---

**Evolved Task: Parity-Constrained Optimization**

**Problem Description:** Sticker counts for 50 students: `[45, 49, 90, 97, 98, 85, 22, 95, 34, 51, 62, 92, 73, 40, 37, ...]` (truncated) You have **10 operations**. Each operation increases or decreases a count by 1.
**New Rules:** 1. You must maximize the frequency of a target number $x$. 2. **Constraint:** The target number $x$ must be an **odd integer**.
**Question:** What is the maximum number of students who can share the same **odd** sticker count?

---

**Analysis.** The evolution scales the search space and adds a property constraint:

- **Search Space:** Moving from $n = 5$ to $n = 50$ makes brute-force approaches infeasible. The solver must sort the array

and use a sliding window approach.
- **Constraint Satisfaction:** The "odd number" constraint requires the solver to filter potential optimal targets, adding a check step often missed by generic "most frequent element" algorithms.

## F.4. Temporal Reasoning: Scheduling

### Seed Task: Interval Scheduling

**Problem Description:** Activity list with `(duration, deadline)`: `[(937, 536), (716, 37), (365, 413), (796, 1813), (31, 1006)]` Residents start from Day 1. You cannot perform two activities simultaneously.
**Question:** What is the maximum number of activities you can complete?

### Evolved Task: Scheduling with Release Times

**Problem Description:** Project list with `(earliest_start, duration, deadline)`: `[(3, 20, 49), (28, 13, 53), (22, 11, 44), (37, 11, 57), (20, 1, 42), (40, 19, 87)]`
**New Rules:** 1. **Release Time:** You cannot start a project before its `earliest_start` time. If the current time is earlier, you must wait (idle). 2. **Non-preemptive:** Once started, a project occupies the timeline for `duration` days continuously. 3. **Deadline:** A project must be completed by its `deadline` (Completion Time ≤ Deadline).
**Question:** What is the maximum number of projects you can complete?

**Analysis.** This evolution transforms a standard greedy problem (Earliest Deadline First) into a more complex variant:

- **Resource Constraint:** The addition of 'earliest_start' (release time) invalidates simple greedy strategies that assume tasks are always available. It forces the model to consider "wait versus process" trade-offs, often requiring dynamic programming or priority-queue simulations.

## F.5. Commonsense Reasoning: Simulation

### Seed Task: 1D Battle Simulation

**Problem Description:** Battlefield is a 1D line of length 10 (Coordinates 0-10). Soldiers: `[(Faction=0, Power=9, Pos=2), (Faction=1, Power=4, Pos=4), ...]` **Rules:** 1. All soldiers move 1 unit/day towards the enemy base (Faction 0 moves right, Faction 1 moves left). 2. **Merge:** Same faction at same pos → Merge (Power sums). 3. **Fight:** Opposite faction at same pos → Fight. Winner = Higher Power. Winner loses 'Power of the loser'. Winner gains +10 Power bonus. Tie = Both die.
**Question:** What is the remaining power of Faction 0 and Faction 1 after the simulation ends?

### Evolved Task: Kinematic Battle Simulation

**Problem Description:** Battlefield length is 200. Tanks: `[(Faction=1, Power=10, Pos=168, Speed=6), (Faction=0, Power=23, Pos=185, Speed=11), ...]` (Total 40 units)
**New Rules:** 1. **Variable Speed:** Each tank moves `Speed` units per time step. 2. **Collision Logic:** Since speeds vary, units can "jump over" each other in discrete steps. - **Definition:** Collision checks must detect if trajectories cross or land on the same point between $t$ and $t + 1$. - Resolution: If crossing occurs, they interact at $t + 1$. 3. **Victory Condition:** Same combat rules (Diff + Bonus).
**Question:** Calculate the final total power stationed at Base A (Left) and Base B (Right).

**Analysis.** The evolved task introduces physics-aware simulation requirements:

- **Kinematics:** Variable speeds mean relative positions change non-linearly. The solver cannot simply "shift" arrays; it must calculate trajectory intersections.
- **Edge Cases:** Fast units overtaking slow units (same faction) or crossing through enemies (opposite faction) without landing on the exact same integer coordinate requires rigorous interval intersection logic.

**F.6. Relational Reasoning: Circuit Timing**

---

**Seed Task: Text Processing (Placeholder)**

**Problem Description:** (Note: The original seed task was a simplified text relationship query. We present the concept here.) Given a set of entities and their direct relationships (A is father of B, B is sister of C...), deduce the relationship between A and C.
**Input:** List of relations. **Output:** Relationship string.

---

**Evolved Task: Digital Circuit Critical Path**

**Problem Description:** An engineer designs a logic circuit with components (Gates) and connections. **Components:** `C1 (OR, Delay: 2ns), C2 (OR, Delay: 2ns), ..., C14 (NAND, Delay: 2ns)` **Connections:** `C1 -> C2, C2 -> C3, C3 -> C4, ...` (DAG structure) **Input Signals:** - C1: Arrives at 7ns - C2: Arrives at 10ns - C3: Arrives at 5ns
**Rules:** 1. Output Time = Max(Input Arrival Times) + Component Delay. 2. Signal propagates from inputs to outputs. 3. **Critical Path:** The maximum delay from any input to any output.
**Question:** What is the critical path delay (in ns) of this circuit?

---

**Analysis.** The evolution moves from static entity-relation lookups to dynamic graph traversal:

- **DAG Processing:** The circuit defines a directed acyclic graph where node values (time) depend on predecessors.
- **Accumulation Logic:** Unlike simple pathfinding, this requires calculating 'max' arrival times at each gate, simulating parallel signal propagation.

**F.7. Causal Reasoning: Event Chain Inference**

---

**Seed Task: Simple Cause-Effect (Placeholder)**

**Problem Description:** Given a set of IF-THEN rules about numbers (If $x > 5$, then $y = x + 1$). **Input:** Initial $x$.
**Output:** Final value of $y$.

---

**Evolved Task: Causal Chain Reconstruction**

**Problem Description:** A series of events and conditional statements. **Events:** A6: Ice Melting, B7: Hospital Overload, C5: AC Failure, D17: Animal Migration, E20: Comm Failure... **Conditions (Evidence):** (1) If [Ice Melting], usually leads to [Hospital Overload]. (2) If NOT [Ice Melting], [Hospital Overload] unlikely. (3) If [Hospital Overload], leads to [AC Failure]. (4) If NOT [Hospital Overload], [AC Failure] unlikely. (5) If [AC Failure], leads to [Animal Migration]. ... (Claims with varying reliability, e.g., "Someone claims X causes Y but evidence is weak").
**Question:** Identify the correct causal chain from initial event to final result, ignoring unreliable evidence. Output the sequence of event IDs (e.g., a6b7c5...).

---

**Analysis.** The evolved task introduces noise and counterfactual logic:

- **Evidence Filtering:** The solver must distinguish between "strong evidence" (Rules 1-8) and "weak claims" (Claims 10-12), filtering out distractors.
- **Chain Chaining:** It requires linking multiple conditional steps ($a \rightarrow b \rightarrow c$) while verifying necessary conditions ("If NOT a, then NOT b").

## G. Training Implementation Details

### G.1. Experimental Design and Comparison Protocols

**Dual-Model Strategy.** To balance attribution analysis and performance ceiling exploration, we adopt a dual-model strategy:

- **Attribution Analysis:** Our primary experiments use **Qwen3-8B-Base**. Choosing the Base model as a starting point aims to *reduce* interference from potential post-training data and alignment strategies inherent in existing Instruct/Thinking models, ensuring observed changes are primarily driven by the current RLVR process and data mixing strategies.
- **Performance Ceiling Exploration:** We supplement our results with training on **Qwen3-8B(Thinking)** and its Thinking variants to verify whether SSLOGIC can still yield gains on top of stronger baselines.

**Fixed Optimization Steps.** We adhere to a *Fixed Optimization Steps* principle. All comparison groups control the global batch size and total update steps to be consistent. We report checkpoints at steps 160, 200, and 240. The 240-step point corresponds to the upper limit of approximately one epoch for our maximum data configuration. Training employs *Group Relative Policy Optimization* (GRPO) with the *K3 estimator* for unbiased KL regularization. We select GRPO to minimize the influence of SFT behavioral cloning, thereby more clearly comparing the role of reward signals provided by different data sources. We explicitly avoid introducing distillation from stronger models to isolate the contribution of the synthesized data itself.

### G.2. Unbiased KL Divergence Estimator

To ensure training stability, we employ an unbiased estimator for the KL divergence term, also known as the K3 estimator (DeepSeek-AI et al., 2025b). Standard KL estimators can exhibit high variance when the policy $\pi_\theta$ significantly deviates from the reference policy $\pi_{\text{ref}}$. The unbiased estimator is defined as:

$$\mathbb{D}_{\text{KL}}^{\text{unbiased}}(\pi_\theta \| \pi_{\text{ref}}) \approx \frac{\pi_\theta(o|q)}{\pi_{\text{old}}(o|q)} \left( \frac{\pi_{\text{ref}}(o|q)}{\pi_\theta(o|q)} - \log \frac{\pi_{\text{ref}}(o|q)}{\pi_\theta(o|q)} - 1 \right) \tag{4}$$

where $\pi_{\text{old}}$ is the policy from the previous iteration. This formulation reduces gradient variance and improves optimization stability during the RLVR process.

### G.3. Group-Level Rejection Sampling

During the rollout phase, we generate $n = 16$ trajectories for each prompt. To improve the quality of the training signal, we apply group-level rejection sampling. Specifically, we filter out trajectories that fail to pass the verifier (i.e., incorrect answers or execution errors) before computing the advantages. This ensures that the policy updates are driven primarily by successful reasoning traces, which is particularly crucial for logic puzzles where the solution space is sparse. If all trajectories for a given prompt fail, the prompt is skipped for the current update step to avoid introducing noise from purely negative samples.

## H. Hyperparameters

We detail the hyperparameters used for data production, training, and evaluation in this section.

### H.1. Data Production Setup

For the Gate 2 consensus used to determine $y_k^*$, we set the validator pool size to $m = 2$, so the consensus aggregates three scorers in total: the primary solver $S(\boldsymbol{x}_{k,\text{state}})$ plus two pool variants $S_{\text{pool}}^{(1)}(\boldsymbol{x}_{k,\text{state}})$ and $S_{\text{pool}}^{(2)}(\boldsymbol{x}_{k,\text{state}})$. For Code-Augmented Blind Review, we use $n = 5$ reviewers and an agreement threshold of $\tau \geq 3$ (i.e., at least 3/5 agreement). The five blind-review instances are sampled with target difficulties $\{3, 5, 5, 7, 7\}$.

### H.2. Training Setup

We train our models on the SSL dataset using the GRPO algorithm. The base model is initialized from Qwen3-8B-Base. We use a learning rate of $2 \times 10^{-6}$ with dynamic batch sizing enabled. To maintain stability, we apply a KL divergence penalty with a coefficient of $\beta = 0.001$, utilizing the *low_var_kl* estimator (§G.2). The maximum prompt length is set to 8192 tokens, and the maximum response length is 16384 tokens. Overlong prompts are filtered out.

For rollout generation, we use vLLM with a sampling temperature of 0.85 and top-p of 1.0, generating $n = 16$ rollouts per prompt. The reward model is based on Qwen3-8B (non-Thinking). We set the global batch size to 128 **prompts**, which results in $128 \times 16 = 2048$ trajectories per iteration. These trajectories are processed using a PPO mini-batch size of 1024. We also employ group-level rejection sampling (§G.3) during training.

## H.3. Evaluation Setup

We evaluate our models using vLLM with the following generation configuration: temperature $t_{\text{temperature}} = 0.6$, top-p $p = 0.95$, top-k $k = 20$, min-p 0, and a maximum token limit of 16384.

We report performance on various benchmarks using the **pass@**$k$ metric (§B).

## I. Paired-Subset, Subsampled, and Bootstrap Analyses

**Aligned Subset Verification.** To mitigate selection bias arising from unequal subset sizes in the main analysis, we perform a strictly paired evaluation on the intersection of question IDs (170 questions derived from 85 seeds). Table 15 presents the results on this aligned set. The trends corroborate our main findings: DeepSeek-evolved tasks demonstrate smoother difficulty scaling (simultaneous gains in pass@1 and pass@8), whereas o4-mini-evolved tasks exhibit a stronger polarization, driving higher sampling gains at the cost of single-shot stability.

**Isolating Dataset Size: Subsampled Results.** To isolate task quality from dataset size, we evaluate Seed and Evolve datasets with identical question counts ($n = 236$, difficulty $d = 7$). Table 14 provides the complete benchmark breakdown across three training stages. The consistent performance gap, even with restricted data quantity, confirms that the structural complexity and algorithmic diversity of the evolved task families are the primary drivers of reasoning improvement.

*Table 14.* **Full Benchmark Results for Subsampled Experiment (**$n = 236$ **versus** 236**).** Results are presented for Seed and Evolve datasets with identical question counts to isolate quality-driven gains.

| Data@Step | Logic Benchmarks | | | | | | Mathematical Benchmarks | | | | | |
|---|---|---|---|---|---|---|---|---|---|---|---|---|
| | SynLogic | ARC | BBH | BBEH | Enigmata | Δ | AIME24 | AIME25 | Brumo | HMMT | Math500 | Δ |
| Seed@160 | 12.0 | 3.1 | 72.7 | 14.0 | 11.7 | – | 11.9 | 10.8 | 21.1 | 2.8 | 77.8 | – |
| Evolve@160 | 13.7 | 3.8 | 74.2 | 14.3 | 12.6 | +1.0 | 12.5 | 11.1 | 22.8 | 2.1 | 79.3 | +0.7 |
| Seed@200 | 12.7 | 1.9 | 73.8 | 14.5 | 11.9 | – | 11.3 | 10.5 | 20.5 | 2.7 | 78.0 | – |
| Evolve@200 | 15.2 | 3.3 | 74.7 | 15.0 | 12.8 | +1.2 | 14.8 | 11.4 | 24.4 | 2.2 | 79.4 | +1.8 |
| Seed@240 | 13.1 | 2.6 | 74.5 | 14.8 | 11.6 | – | 13.1 | 10.3 | 20.0 | 2.8 | 79.5 | – |
| Evolve@240 | 13.9 | 4.1 | 72.8 | 14.6 | 12.7 | +0.3 | 13.5 | 10.7 | 24.1 | 3.2 | 79.0 | +1.0 |

*Table 15.* **Aligned Subset Results (**$n = 170$**).** Comparisons are strictly paired by Question ID to ensure test set uniformity.

| Generator | Solver | Data | pass@1 | pass@8 |
|---|---|---|---|---|
| DS | DS | Seed | 0.568 | 0.659 |
| | | Evolve | **0.610** | **0.753** |
| DS | DB | Seed | **0.606** | **0.824** |
| | | Evolve | 0.590 | 0.706 |
| o4-mini | DS | Seed | 0.574 | 0.694 |
| | | Evolve | **0.613** | **0.765** |
| o4-mini | DB | Seed | 0.578 | 0.765 |
| | | Evolve | **0.594** | **0.812** |

**Bootstrap Confidence Intervals.** To assess the statistical stability of the observed shifts, we compute 95% confidence intervals for the performance deltas (Δ = Evolve − Seed) using 2,000 bootstrap iterations. As shown in Table 16, although the intervals frequently cross zero due to the limited sample size ($n = 170$), the directional trends are consistent.

*Table 16.* **Bootstrap Analysis of Evolution Effects.** Intervals represent the 95% CI of the delta (Evolve − Seed).

| Generator | Solver | Δpass@1 (95% CI) | Δpass@8 (95% CI) |
|---|---|---|---|
| DS | DS | 0.067 [−0.03, 0.16] | 0.080 [−0.01, 0.18] |
| DS | DB | 0.026 [−0.06, 0.11] | −0.025 [−0.12, 0.06] |
| o4-mini | DS | −0.008 [−0.12, 0.10] | 0.026 [−0.09, 0.13] |
| o4-mini | DB | −0.018 [−0.13, 0.09] | 0.026 [−0.08, 0.12] |

# J. Sampling, Filtering, and Token-Metric Details

**Shuffle and filtering protocol.** We shuffle tasks within each source before sampling up to 100 tasks. Rejected samples (ill-formed prompts, validator failures, or parse errors) are discarded; we rerun generation up to five times to replace missing samples. All statistics in § 5.1 and § 5.2 are computed on accepted tasks only.

**Reflection-like Token Analysis.** Following the methodology of DeepSeek-R1 (DeepSeek-AI et al., 2025a), we define the following vocabulary as proxy signals for "reflection-like" behaviors: *wait*, *mistake*, *however*, *but*, *retry*, *error*, *verify*, *wrong*, *evaluate*, *check*. We compute the raw frequency of these tokens on the fixed validation set using identical decoding settings across all checkpoints. This allows us to characterize the emergence of CoT-style self-correction markers independently of response length changes.

*Table 17.* **Filtering summary for analysis samples.** Parse% reports the fraction of samples removed due to parsing or validation failures (Seed excludes one known problematic sample); summary statistics are computed on accepted tasks only.

| Source | Parse% | Lines of Code | Cyclomatic | Cognitive | Halstead Volume | Halstead Effort | Loop Depth | Loops |
|---|---|---|---|---|---|---|---|---|
| seed | 1.0% | 136.1 | 27.96 | 49.41 | 5249.9 | 174348.7 | 2.07 | 10.33 |
| deepseek | 1.0% | 191.8 | 79.77 | 147.32 | 7734.8 | 597193.1 | 2.42 | 22.15 |
| o4-mini | 1.0% | 121.6 | 57.04 | 91.83 | 5274.6 | 341555.1 | 2.35 | 19.91 |
| glm | 2.4% | 203.3 | 93.67 | 165.12 | 8802.7 | 670114.5 | 2.65 | 27.61 |

*Table 18.* **Difficulty-level solvability values.** Pass@1 and pass@8 values for Figure 6, using the DeepSeek single-round evolution set and the two reference solvers (DeepSeek-V3.1-Terminus and Doubao-1.6-Thinking).

| Metric | Data | Solver | $d=5$ | $d=7$ | $d=10$ |
|---|---|---|---|---|---|
| pass@1 | Seed | DeepSeek | 0.6727 | 0.6000 | 0.5660 |
| | Seed | Doubao | 0.7091 | 0.5818 | 0.5094 |
| | Evolve | DeepSeek | 0.7091 | 0.6909 | 0.5660 |
| | Evolve | Doubao | 0.7273 | 0.6727 | 0.4906 |
| pass@8 | Seed | DeepSeek | 0.7818 | 0.6364 | 0.6981 |
| | Seed | Doubao | 0.8364 | 0.7818 | 0.7925 |
| | Evolve | DeepSeek | 0.8545 | 0.8000 | 0.6981 |
| | Evolve | Doubao | 0.8545 | 0.7818 | 0.6981 |

# K. Complexity Metric Definitions

## K.1. Metric Definitions and Computation

All complexity metrics are computed from the Python AST of each paired generator–validator program. We summarize each metric below.

**Lines of Code (LOC).** Total lines are split into blank lines, comment lines (starting with #), and docstring lines. LOC reports only non-empty, non-comment, non-docstring lines.

**Cyclomatic Complexity.** Starts at 1 and increments by one for each decision point (*if*, *for*, *while*, *try/except*, *with*, boolean operators, and comprehensions).

**Cognitive Complexity.** Adds 1 for each control structure and adds a nesting penalty proportional to the current nesting depth (e.g., nested *if*/*for*/*while*/*try* blocks increase the score).

**Halstead Metrics.** Operators and operands are counted from the AST to compute vocabulary $n$, length $\ell$, volume $v = \ell \log_2 n$, difficulty $d = (n_1/2) \cdot (\ell_2/n_2)$, and effort $e = d \times v$.

**Loop Depth and Loop Count.** Loop count sums all *for*, *while*, and comprehension loops; loop depth tracks the maximum nesting level of loops.

**Recursion.** A function is marked recursive if its call graph contains a self-edge (function calls itself).

**Algorithmic Patterns.** Pattern detectors are heuristic and non-exclusive. We identify patterns using syntactic markers. **Binary Search:** flagged by *while* loops containing tokens such as *mid*, *left*, *right*, *low*, or *high*. **Sorting:** detected by calls to *sort*, *sorted*, *heapify*, *heappush*, or *heappop*. **Dynamic Programming:** identified by variable names containing *dp*, *memo*, *cache*, or *tabulation*, or by the use of decorators like *@lru_cache* or *@cache*. **Graph Traversal:** flagged by identifiers such as *bfs*, *dfs*, *dijkstra*, *bellman*, *floyd*, *graph*, *visited*, *queue*, or *stack*. **Divide-and-Conquer:** indicated by the presence of recursive calls alongside tokens such as *mid*, *half*, or *divide*.

**Inferred Time Complexity.** A rule-based heuristic maps the detected structure to complexity buckets. Unmatched structures are recorded as *Other*. Buckets include Constant/Logarithmic ($O(1)$, $O(\log n)$), Polynomial ($O(n)$, $O(n \log n)$), $O(n^2)$, $O(n^3)$, $O(n^4+)$), Graph/Multivariate ($O(nm)$, $O(|V|+|E|)$, $O(|V|^2)$, $O(|V|+|E|)/O(|V|^2)$), and Exponential ($O(2^n)$).

## L. Human Audit Protocol

To assess the quality of synthesized task families beyond automated metrics, we conducted an independent human audit of 100 randomly sampled Evolve families generated by o4-mini. For each family, the complete pipeline artifacts—Generator, Ground Truth Validator, two independently re-derived Auto-Validators, Question Template, and a concrete sample instance (question + answer)—were reviewed across four dimensions.

*Table 19.* **Human audit results** on 100 Evolve task families. Each dimension is graded A/B/C/D; Pass Rate reports A+B.

| Dimension | A | B | C | D | Pass (A+B) |
|---|---|---|---|---|---|
| Validator Consistency | 92 | 1 | 0 | 7 | 93% |
| Answer Correctness | 89 | 1 | 3 | 7 | 90% |
| Question Clarity | 75 | 19 | 3 | 3 | 94% |
| Generator Quality | 81 | 13 | 2 | 4 | 94% |

Overall, **85 out of 100 families are usable** (78 fully correct, 7 with minor issues). Among the remaining 15 failure cases:

- **Validator Logic Disagreement (5 cases):** The ground truth validator and auto-validators implement fundamentally different solving logic—e.g., different bracket matching semantics, incorrect greedy strategies, or LeetCode-specific logic contradicting the prompt. The triple-validator consensus mechanism successfully *flags* these cases.
- **Prompt–Code Semantic Mismatch (5 cases):** All three validators agree internally, but the implemented logic contradicts the natural language description—e.g., bidirectional swaps described but directional constraints enforced. This is the dominant undetected failure mode.
- **Algorithmic Correctness Issues (3 cases):** All validators agree but implement a flawed algorithm that passes simple test cases but fails on harder inputs.
- **Missing Information (2 cases):** Insufficient specification in the prompt leading to ambiguous interpretation.

The dominant failure pattern is prompt–code semantic mismatch, where the evolution preserves algorithmic logic from well-known problems while altering the natural language description, creating a semantic gap. This represents a clear direction for future improvement in the synthesis pipeline.

## M. Enigmata Experiment Details

This appendix provides the full protocol for the size-controlled Enigmata experiment described in §4.5.

### M.1. Seed Selection and Evolution Protocol

We started from the 36 puzzle types in the public Enigmata benchmark (Chen et al., 2025a). Six types were excluded because they lack parameterized difficulty control (i.e., their generators do not accept a difficulty parameter $d$), leaving **30 types** for evolution.

For each type, we applied our Evolve pipeline using **GPT-5.4** as the synthesis agent, with $t_{\max} = 2$ refinement iterations and the same Multi-Gate Validation Protocol described in §3. The evolution agent receives the original Enigmata generator–

validator pair as seed and produces a structurally modified variant.

## M.2. Instance Generation and Matching

After evolution, instances were generated at difficulty levels $d \in \{5, 7, 10\}$ following the same distribution as the original Enigmata data. To ensure a fair comparison, we matched instance counts as closely as possible:

*Table 20.* **Enigmata experiment configuration.**

|  | **Enigmata-Seed** | **Enigmata-Evolve** |
| --- | --- | --- |
| Puzzle types | 30 | 30 |
| Total instances | 8,776 | 8,758 |
| Difficulty levels | $\{5, 7, 10\}$ | $\{5, 7, 10\}$ |
| Synthesis agent | — | GPT-5.4 |
| Refinement iterations | — | 2 |

## M.3. Training Configuration

Both Seed and Evolve models were trained on **Qwen3-8B-Base** using GRPO for 240 steps, with identical hyperparameters (learning rate, batch size, KL penalty) as the main experiment (Appendix H). No other data was mixed in—each model was trained purely on its respective Enigmata variant.

## M.4. Key Properties

This experiment provides three important controls:

1. **Seed-agnostic:** The Enigmata seeds share no overlap with our original 400 Chinese seed families, demonstrating the pipeline generalizes beyond its original seed distribution.
2. **Zero human annotation:** The entire pipeline—from seed ingestion through evolution to training—requires no human intervention beyond selecting the 30 eligible types.
3. **English-only:** Unlike our main experiment where seeds are in Chinese, Enigmata is entirely English, confirming cross-language applicability.

# N. Token-Matched Analysis

Step-matched comparisons may not fully control for compute, as Evolve tasks elicit progressively longer reasoning chains. This appendix provides a token-budget analysis using per-step training logs.

## N.1. Token Consumption

At Step 240, Evolve consumes $\sim 1.20\times$ the cumulative tokens of Seed (1.299B vs 1.087B). This gap arises because Evolve tasks elicit progressively longer responses ($\sim 2,150 \rightarrow \sim 3,300$ tokens) while Seed triggers shorter ones ($\sim 2,400 \rightarrow \sim 1,200$ tokens). Figure 9 shows the cumulative token curves for all five experiments.

## N.2. Token-Matched Comparison

To isolate per-token training value from the additional compute, we compare at matched cumulative token budgets:

- **Seed at Step 240** (1.087B tokens): Macro Accuracy = 0.2312
- **Evolve at** $\sim$**Step 210** ($\sim$1.087B tokens): Macro Accuracy = 0.2480

After matching the total RL token budget, Evolve still outperforms Seed by **+7.3% relative**. This indicates that approximately half of the step-matched gap (+14.2% relative) comes from higher per-token training value, with the remainder attributable to additional compute via longer responses.

## N.3. Response Length Dynamics

Figure 11 shows the mean response length evolution during training. Evolve's responses grow steadily, reflecting the emergence of more detailed reasoning chains, while Seed's responses plateau or decrease.

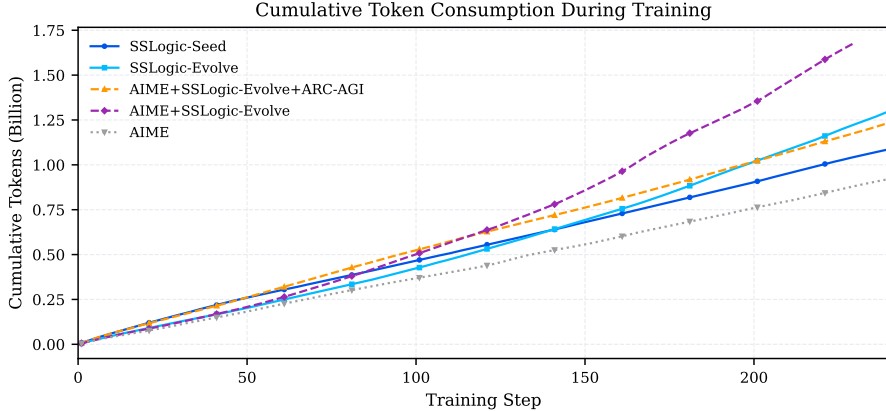

*Figure 9.* Cumulative RL token consumption across experiments.

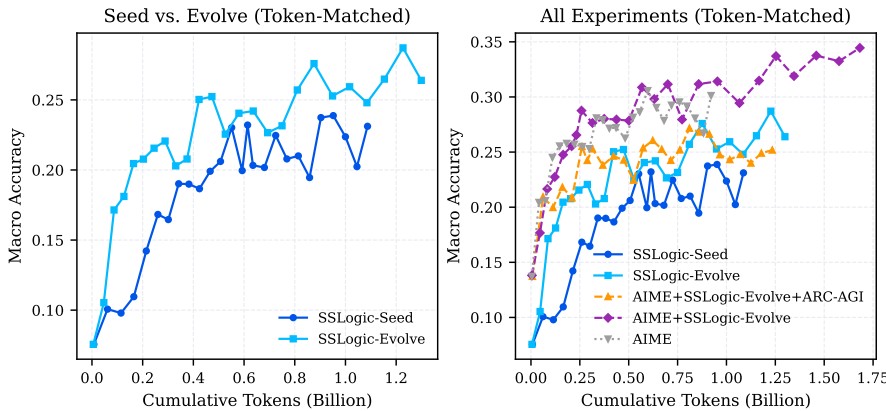

*Figure 10.* Macro Accuracy vs. cumulative RL tokens. At matched token budgets (~1.09B), Evolve maintains a clear advantage.

### N.4. Token Efficiency

Figure 12 provides a per-token efficiency comparison, plotting accuracy gain per billion tokens consumed for each experiment.

## O. Prompt Templates

### O.1. Cognitive Kernel Pro (CKPro) Framework Prompts

We adopt the Cognitive Kernel Pro (CKPro) framework (Fang et al., 2025) as the underlying agent architecture. CKPro utilizes a structured prompt system to manage task planning, action execution, final output formatting, and result aggregation. The core prompts are detailed below.

#### O.1.1. PLANNING PROMPT

The Planning module is responsible for maintaining the high-level progress state and generating the next strategic plan.

---

**CKPro Planning System Prompt**

You are a strategic assistant responsible for the high-level planning module of the Cognitive Kernel, an initial autopilot system designed to accomplish user tasks efficiently.
**Available Information**

- `Target Task`: The specific task to be completed.

---

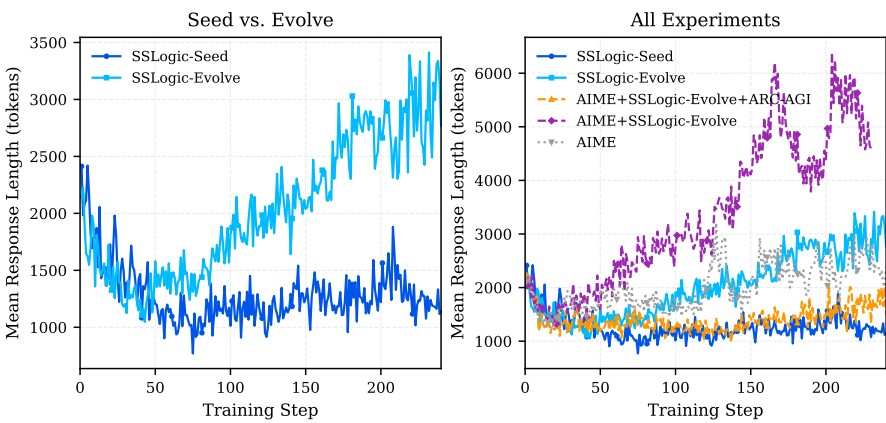

*Figure 11.* Mean response length during training.

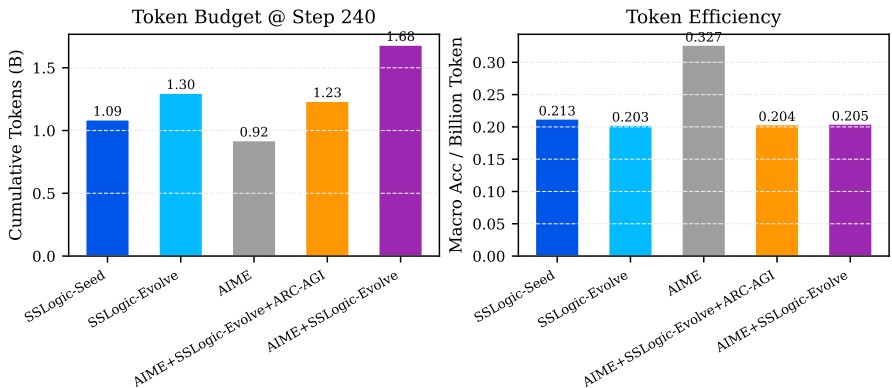

*Figure 12.* Token efficiency: accuracy gain per billion cumulative tokens.

- `Recent Steps`: The most recent actions taken by the agent.
- `Previous Progress State`: A JSON representation of the task's progress, including key information and milestones.
- `Sub-Agent Functions` and `Tool Functions`: Definitions of available sub-agents and tools for task execution.

**Progress State**

The progress state is crucial for tracking the task's advancement and includes:

- `completed_list` (`List[str]`): A list of completed steps and gathered information essential for achieving the final goal.
- `todo_list` (`List[str]`): A list of planned future steps; aim to plan multiple steps ahead when possible.
- `experience` (`List[str]`): Summaries of past experiences and notes, such as failed attempts or special tips, to inform future actions.
- `information` (`List[str]`): A list of collected important information from previous steps. These records serve as the memory and are important for tasks such as counting (to avoid redundancy).

Here is an example progress state for a task to locate and download a specific paper for analysis:

```
{
  "completed_list":  ["Located and downloaded the paper (as 'paper.pdf')
using the web agent.", "Analyze the paper with the document agent."], #
completed steps
  "todo_list":  ["Perform web search with the key words identified from the
paper."], # todo list
  "experience":  [], # record special notes and tips
  "information":  ["The required key words from the paper are AI and NLP."],
# previous important information
}
```

**Guidelines**

1. **Objective**: Update the progress state and adjust plans based on previous outcomes.

2. **Code Generation**: Create a Python dictionary representing the updated state. Ensure it is directly evaluable using the `eval` function. Check the `Progress State` section above for the required content and format for this dictionary. Keep every string value on a single line—do not embed raw newline characters inside string literals.

3. **Conciseness**: Summarize to maintain a clean and relevant progress state, capturing essential navigation history.

4. **Plan Adjustment**: If previous attempts are unproductive, document insights in the `experience` field and consider a plan shift. Nevertheless, notice that you should NOT switch plans too frequently.

5. **Utilize Resources**: Effectively employ sub-agents and tools to address sub-tasks.

**Output Format**

**CRITICAL: You MUST output EXACTLY ONE response with ONE Thought and ONE Code block. DO NOT output multiple Thought-Code pairs. DO NOT continue thinking after the Code block.**

Your response should strictly follow this format:

```
Thought:  {Provide your reasoning and explanation}
Code:  {Output your Python dictionary wrapped in ```python ``` marks}
```

**After outputting the Code block, STOP IMMEDIATELY. Do Not add any additional Thought, Observation, or Code blocks.**

**Strategies**

1. **Be Meticulous and Persistent**:
   - Carefully inspect every stage of your process, and re-examine your results if you notice anything unclear or questionable.
   - Stay determined – don't give up easily. If one strategy does not succeed, actively seek out and try different approaches.

2. **Task Decomposition and Execution**:
   - **Break Down the Problem**: Divide complex tasks into clear, self-contained sub-tasks. Each sub-task description should include all necessary information, as sub-agents (or tools) do not have access to the full context.
   - **Sequential Processing**: Address each sub-task one at a time, typically invoking only one sub-agent (or tool) per step. Review results before proceeding to minimize error propagation.
   - **Stable Sub-agent Use**: Treat sub-agents (or tools) as independent helpers. Ensure that each sub-task is well-defined and that input/output types are compatible.
   - **Direct LLM Use**: If the remaining problem can be solved by a language model alone (e.g., requires reasoning but no external data), use `ask_llm` to complete the task.

3. **Adaptive Error Handling and Result Integration**:
   - **Monitor and Reflect**: After each step, carefully review the outcome – including any errors, partial results, or unexpected patterns. Use this information to decide whether to retry, switch to an alternative method, or leverage partial results for the next action.
   - **Limited Intelligent Retrying**: If the error appears transient or recoverable (e.g., network issues, ambiguous queries), retry the step once (for a total of two attempts). If the error persists after the retry, do not continue; proceed to an alternative method or tool.
   - **Alternative Strategies**: If both attempts fail or the error seems fundamental (e.g., tool limitations, unavailable data), switch to an alternative approach to achieve the sub-task's goal.
   - **Partial Result Utilization**: Even if a sub-task is not fully completed, examine any partial results or error messages. Use these to inform your next steps; partial data or observed error patterns can guide further actions or suggest new approaches.
   - **Leverage Existing Results**: Access results from the Progress State or Recent Steps sections, and use any previously downloaded files in your workspace.
     - Avoid writing new code to process results if you can handle them directly.
     - Do not assume temporary variables from previous code blocks are still available.
   - **Prevent Error Propagation**: By handling one sub-task at a time, reviewing outputs, and adapting based on feedback, you reduce the risk of compounding errors.

4. **Multi-agent Collaboration Patterns**:
   - **Step-by-Step Coordination**: When handling complex tasks, coordinate multiple specialized sub-agents (tools) in a step-by-step workflow. To minimize error propagation, use only one sub-agent or tool per step, obtaining its result before proceeding to the next.
   - **General Guidelines**:
     - **Use sub-agents as modular helpers**: Each sub-agent is already defined and implemented as a function with clearly defined input and output types.
     - **Review Definitions**: Carefully review the definitions and documentation strings of each sub-agent and tool in the `Sub-Agent Function` and `Tool Function` sections to understand their use cases. Do not re-define these functions; they are already provided.
     - **Explicitly Specify Requirements**: Sub-agents operate independently and do not share context or access external information. Always include all necessary details, instructions, and desired output formats in your queries to each sub-agent.
     - **Define Output Formats**: Clearly state the required output format when requesting information to ensure consistency and facilitate downstream processing.
   - **Typical Workflows**:
     - Example 1, Analyzing a File from the Web: (1) Use `simple_web_search` to find the file's URL (this step can be

optional but might usually be helpful to quickly identify the information source). (2) Use `web_agent` to download the file using the obtained URL (note that `web_agent` usually cannot access local files). (3) Use `file_agent` to process the downloaded file.
  – Example 2, Finding Related Information for a Keyword in a Local File: (1) Use `file_agent` to analyze the file and locate the keyword. (2) Use `simple_web_search` to search for related information. (3) Use `web_agent` to gather more detailed information as needed.
  – Example 3, Reading Simple Text Files: (1) For simple text files (e.g., `.txt`, `.json`, `.md`, `.yaml`), use `read_text` directly to get the content quickly. (2) For complex file types (PDF, Excel, images) or when advanced processing is needed, use `file_agent`.
  – Complex Tasks: For more complex scenarios, you may need to interleave calls to different sub-agents and tools. Always specify a clear, step-by-step plan.
- **Important Notes**:
  – Each sub-agent call is independent; once a call returns, its state is discarded.
  – The only channels for sharing information are the input and output of each sub-agent call (and the local file system).
  – Maximize the information provided in the input and output to ensure effective communication between steps.

*Figure 13.* The system prompt used for the planning module in the Cognitive Kernel Pro framework.

## O.1.2. ACTION PROMPT

The Action module generates the specific Python code to execute the next step defined by the planner.

---

### CKPro Action System Prompt

You are a strategic assistant responsible for the action module of the Cognitive Kernel, an initial autopilot system designed to accomplish user tasks. Your role is to generate a Python code snippet to execute the next action effectively.
**Available Information**

- `Target Task`: The specific task you need to complete.
- `Recent Steps`: The most recent actions you have taken.
- `Progress State`: A JSON representation of the task's progress, including key information and milestones.
- `Sub-Agent Functions` and `Tool Functions`: Definitions of available sub-agents and tools for use in your action code.

**Coding Guidelines**

1. **Output Management**: Use Python's built-in `print` function to display results. Printed outputs are used in subsequent steps, so keep them concise and focused on the most relevant information.
2. **Self-Contained Code**: Ensure your code is fully executable without requiring user input. Avoid interactive functions like `input()` to maintain automation and reproducibility.
3. **Utilizing Resources**: Leverage the provided sub-agents and tools, which are essentially Python functions you can call within your code. Notice that these functions are **already defined and imported** and you should NOT re-define or re-import them.
4. **Task Completion**: Use the `stop` function to return a well-formatted output when the task is completed.
5. **Python Environment**: Explicitly import any libraries you need, including standard ones such as `os` or `sys`, as nothing (except for the pre-defined sub-agents and tools) is imported by default. You do NOT have sudo privileges, so avoid any commands or operations requiring elevated permissions.
6. **Working Directory**: Use the current folder as your working directory for reading from or writing to files.
7. **Complexity Control**: Keep your code straightforward and avoid unnecessary complexity, especially when calling tools or sub-agents. Write code that is easy to follow and less prone to errors or exceptions.

**Output Format**
**CRITICAL: You MUST output EXACTLY ONE response with ONE Thought and ONE Code block. DO NOT output multiple Thought-Code pairs. DO NOT continue thinking after the Code block.**
Your response should strictly follow this format:
`Thought:  {Provide your reasoning and explanation}`
`Code:  {Output your executable Python code wrapped in ```python ``` marks}`
**After outputting the Code block, STOP IMMEDIATELY. Do NOT add any additional Thought, Observation, or Code blocks.**
**Strategies**

1. **Be Meticulous and Persistent**:
   - Carefully inspect every stage of your process, and re-examine your results if you notice anything unclear or questionable.
   - Stay determined – don't give up easily. If one strategy does not succeed, actively seek out and try different approaches.

2. **Task Decomposition and Execution**:
   - **Break Down the Problem**: Divide complex tasks into clear, self-contained sub-tasks. Each sub-task description should include all necessary information, as sub-agents (or tools) do not have access to the full context.
   - **Sequential Processing**: Address each sub-task one at a time, typically invoking only one sub-agent (or tool) per step. Review results before proceeding to minimize error propagation.
   - **Stable Sub-agent Use**: Treat sub-agents (or tools) as independent helpers. Ensure that each sub-task is well-defined and that input/output types are compatible.
   - **Direct LLM Use**: If the remaining problem can be solved by a language model alone (e.g., requires reasoning but no external data), use `ask_llm` to complete the task.
3. **Adaptive Error Handling and Result Integration**:
   - **Monitor and Reflect**: After each step, carefully review the outcome – including any errors, partial results, or unexpected patterns. Use this information to decide whether to retry, switch to an alternative method, or leverage partial results for the next action.
   - **Limited Intelligent Retrying**: If the error appears transient or recoverable (e.g., network issues, ambiguous queries), retry the step once (for a total of two attempts). If the error persists after the retry, do not continue; proceed to an alternative method or tool.
   - **Alternative Strategies**: If both attempts fail or the error seems fundamental (e.g., tool limitations, unavailable data), switch to an alternative approach to achieve the sub-task's goal.
   - **Partial Result Utilization**: Even if a sub-task is not fully completed, examine any partial results or error messages. Use these to inform your next steps; partial data or observed error patterns can guide further actions or suggest new approaches.
   - **Leverage Existing Results**: Access results from the Progress State or Recent Steps sections, and use any previously downloaded files in your workspace.
     - Avoid writing new code to process results if you can handle them directly.
     - Do not assume temporary variables from previous code blocks are still available.
   - **Prevent Error Propagation**: By handling one sub-task at a time, reviewing outputs, and adapting based on feedback, you reduce the risk of compounding errors.
4. **Multi-agent Collaboration Patterns**:
   - **Step-by-Step Coordination**: When handling complex tasks, coordinate multiple specialized sub-agents (tools) in a step-by-step workflow. To minimize error propagation, use only one sub-agent or tool per step, obtaining its result before proceeding to the next.
   - **General Guidelines**:
     - **Use sub-agents as modular helpers**: Each sub-agent is already defined and implemented as a function with clearly defined input and output types.
     - **Review Definitions**: Carefully review the definitions and documentation strings of each sub-agent and tool in the `Sub-Agent Function` and `Tool Function` sections to understand their use cases. Do not re-define these functions; they are already provided.
     - **Explicitly Specify Requirements**: Sub-agents operate independently and do not share context or access external information. Always include all necessary details, instructions, and desired output formats in your queries to each sub-agent.
     - **Define Output Formats**: Clearly state the required output format when requesting information to ensure consistency and facilitate downstream processing.
   - **Typical Workflows**:
     - Example 1, Analyzing a File from the Web: (1) Use `simple_web_search` to find the file's URL (this step can be optional but might usually be helpful to quickly identify the information source). (2) Use `web_agent` to download the file using the obtained URL (note that `web_agent` usually cannot access local files). (3) Use `file_agent` to process the downloaded file.
     - Example 2, Finding Related Information for a Keyword in a Local File: (1) Use `file_agent` to analyze the file and locate the keyword. (2) Use `simple_web_search` to search for related information. (3) Use `web_agent` to gather more detailed information as needed.
     - Example 3, Reading Simple Text Files: (1) For simple text files (e.g., `.txt`, `.json`, `.md`, `.yaml`), use `read_text` directly to get the content quickly. (2) For complex file types (PDF, Excel, images) or when advanced processing is needed, use `file_agent`.
     - Complex Tasks: For more complex scenarios, you may need to interleave calls to different sub-agents and tools. Always specify a clear, step-by-step plan.
   - **Important Notes**:
     - Each sub-agent call is independent; once a call returns, its state is discarded.
     - The only channels for sharing information are the input and output of each sub-agent call (and the local file system).
     - Maximize the information provided in the input and output to ensure effective communication between steps.

**Example**

**Task:** Summarize a random paper about LLM research from the Web

**Step 1**

```
Thought:  Begin by searching the web for recent research papers related to large
```

```
language models (LLMs).
Code:

    ```python
    search_query = "latest research paper on large language models"
    result = simple_web_search(search_query)
    print(result)
    ```
```

**Step 2**

```
Thought:  From the search results, choose a random relevant paper.  Use web_agent to
download the PDF version of the selected paper.
Code:

    ```python
    print(web_agent(task="Download the PDF of the arXiv paper 'Large Language
    Models:  A Survey' and save it as './LLM_paper.pdf'"))
    ```
```

**Step 3**

```
Thought:  With the paper downloaded, use file_agent to generate a summary of its
contents.
Code:

    ```python
    result=file_agent(task="Summarize the paper",
    file_path_dict={"./LLM_paper.pdf":  "Large Language Models:  A Survey"})
    print(result)
    ```
```

**Note**

- Each step should be executed sequentially, generating and running the code for one step at a time.
- Ensure that the action codes for each step are produced and executed independently, not all at once.

*Figure 14.* The system prompt used for the action module in the Cognitive Kernel Pro framework.

O.1.3. FINAL OUTPUT PROMPT

The End module formats the final result when the agent finishes executing the task.

---

**CKPro Final Output System Prompt**

You are a proficient assistant tasked with generating a well-formatted output for the execution of a specific task by an agent.
**Available Information**

- `Target Task`: The specific task to be accomplished.
- `Recent Steps`: The latest actions taken by the agent.
- `Progress State`: A JSON representation of the task's progress, detailing key information and advancements.
- `Final Step`: The last action before the agent's execution concludes.
- `Stop Reason`: The reason for stopping. If the task is considered complete, this will be "Normal Ending".
- `Result of Direct ask_llm` (Optional): For the case where the task is likely to be incomplete, we have an alternative response by directly asking a stand-alone LLM.
**Guidelines**

1. **Goal**: Deliver a well-formatted output. Adhere to any specific format if outlined in the task instructions.
2. **Code**: Generate a Python dictionary representing the final output. It should include two fields: `output` and `log`. The `output` field should contain the well-formatted final output result, while the `log` field should summarize the navigation trajectory.

3. **Final Result**: Carefully examine the outputs from the previous steps as well as the alternative result (if existing) to decide the final output.
4. **Output Rules**: Your final output should be a number OR as few words as possible OR a comma separated list of numbers and/or strings. Do NOT include any unnecessary information in the output.

**Output Format**

**CRITICAL: You MUST output EXACTLY ONE response with ONE Thought and ONE Code block. DO NOT output multiple Thought-Code pairs. DO NOT continue thinking after the Code block.**

Your response should strictly follow this format:
```
Thought:  {Provide your reasoning and explanation}
Code:  {Output your Python dictionary wrapped in ```python ``` marks}
```

**After outputting the Code block, STOP IMMEDIATELY. Do NOT add any additional Thought, Observation, or Code blocks.**

**Output Rules**

- **Number**: If you are asked for a number, directly output the number itself. Don't use comma to write your number. Be careful about what the question is asking; for example, the query might ask "how many thousands". In this case, you should properly convert the number if needed. Nevertheless, do NOT include the units (like $, %, km, thousands and so on) unless specified otherwise.
- **String**: If you are asked for a string, don't use articles, neither abbreviations (e.g., for cities), and write the digits in plain text unless specified otherwise.
- **List**: If you are asked for a comma separated list, apply the above rules depending on whether the element to be put in the list is a number or a string.

**Examples**

Here are some example outputs:

**Example 1**
```
Thought:  The task is completed with the requested price found and I should directly
output the price.
Code:
```

```python
{
  "output":  "799", # provide a well-formatted output
  "log":  "The task is completed.  The result is found by first using the
web_agent to obtain the information and then using Python for calculation.",
# a summary of the navigation details
}
```

**Example 2**
```
Thought:  The task is incomplete with the problem of exceeding max steps, and I
choose to trust the results of direct ask_llm.
Code:
```

```python
{
  "output":  "799",
  "log":  "The alternative result by directly asking an LLM is adopted since
our main problem-solving procedure was incomplete.",
}
```

*Figure 15.* The system prompt used for final output formatting in the Cognitive Kernel Pro framework.

### O.1.4. RESULT AGGREGATION PROMPT

The Aggregation module selects the best result from multiple execution candidates (used in multi-path reasoning or repeated attempts).

---

**CKPro Aggregation System Prompt**

You are a highly capable assistant responsible for selecting the most likely correct result from a list of candidate outputs generated for a specific step in solving a target task.
**Available Information**

- `Target Task`: The specific task to be accomplished.
- `Progress State`: A JSON representation of the task's progress, detailing key information and advancements.
- `Current Step`: The reasoning and actions (executed code) taken at this step.
- `Results to Select`: A list of candidate results produced for the current step.

**Guidelines**

1. **Contextual Review**: Carefully review the `Progress State` and `Current Step` to understand the context and requirements for this selection.
2. **Majority Voting**: By default, select the result that is most consistent with the majority of other results. If multiple results are similar, prefer the one that aligns with the consensus.
3. **Error Exclusion**: Exclude any results that are clearly unreasonable, such as those containing errors, irrelevant information, or signs of failed execution.
4. **Tie-Breaking**: If there is a tie among reasonable results, select the one that is best formatted and provides the most detailed and complete answer.
5. **Fallback**: If none of the results are clearly correct, select the one that appears most reasonable given the context.
6. **Output Format**: Output the index of the selected result using the `print` function. For example, to select the result at index 2, output in your code section: `print(2)`.

---

*Figure 16.* The system prompt used for result aggregation in the Cognitive Kernel Pro framework.

## O.2. Scale Scaling Logic (SSL) Pipeline Prompts

The SSL pipeline for data generation consists of three main stages: **Problem Evolution**, **Quality Verification**, and **Solution Generation**. Additionally, we employ an **Experience Manager** to curate high-quality reasoning strategies and a **Validator Builder** to construct independent validators for voting.

### O.2.1. STAGE 1: PROBLEM EVOLUTION

The core of our pipeline is the **Code Reasoning Agent**, evolving seed problems into complex tasks via three components: a **generator**, a **question_template**, and a **validator**.

---

**Evolution System Prompt (English)**

You are a Code Reasoning Agent responsible for **evolving high-quality logical reasoning problems**.
**Core Task**
Based on the original seed, design **three independent components** to construct a deeper logical reasoning problem, so the reasoning chain undergoes substantive evolution and difficulty increases, rather than merely changing the story surface:
1. **generator**: A Python function `input(difficulty)` that generates input data and slot texts.
    - Return format: `(inputs, slot_texts)`
    - `inputs`: input data passed to the validator
    - `slot_texts`: text list used to fill the question template
2. **question_template**: A text template containing placeholders like `[Input Slot 1]` and `[Input Slot 2]`.
    - Example: `"Given [Input Slot 1] and [Input Slot 2], find..."`
    - Placeholders will be filled by the generator's `slot_texts` in order
3. **validator**: A Python function `solution(inputs)` that computes the correct answer.
    - Accepts `inputs` produced by the generator
    - Returns the standard answer; must be a pure function with no side effects, so the system can auto-generate multiple independent validators for consistency checks

**Delivery Format**

- Use `stop` to output JSON: `task_id`, `generator_code`, `question_template`, `validator_code`, `evolution_strategy`, `sample_summary`, `blind_review_summary`, `experience_updates`, `notes`.
- **Important**: the parameter to `stop` must be a valid JSON string containing all fields above. The system parses this JSON in the final step (end stage); if the output is `None` or cannot be parsed, the task fails.

**Mandatory Constraints**

---

- **Reasoning Evolution First**: The evolution strategy must focus on innovations/deepening in reasoning logic, constraints, or variable relationships. You may reference `mutation_hint`, but do not merely reskin the story or replace terms. Avoid problems that look complex but are actually easy.
- **High Difficulty Target**: Our goal is for top-tier reasoning models to have a high error rate at difficulty 10. Problems must build long reasoning chains, interdependent constraints, or counterintuitive settings; do not intentionally simplify or reduce difficulty.
- **Keep Task Lineage**: The core objective must follow the original seed (see sample question and `task_path` below). You may add new constraints or change the narrative background, but do not turn the problem into an unrelated type; the final answer format must still align with the original task requirements.
- **All Three Components Must Evolve**: `generator_code` (must be rewritten), `question_template` (new statement), `validator_code` (structure can be retained but must match new rules).
- **Question Self-check**: Before blind review, the system calls `seed_check_question_quality()` to check readability, novelty, and difficulty. If it fails, blind review is blocked.
- **Validator Pool Consistency**: After saving, the system auto-generates 2 independent validators based on the generator/template spec. A majority vote becomes the standard answer. If they disagree, fix the main validator and record the experience.
- **Your Code May Be Wrong**: Both generator and validator may contain bugs. If blind review fails, first verify your generator outputs and validator computations in Python before blaming the blind-review LLM.
- **Answer Self-proof**: Do not rely on the seed answer. Use prints/assertions/small Python scripts to repeatedly verify solvability, correctness, and validator judgments.
- **Single-step Tooling**: Keep operations atomic; do not call tools inside `stop`.
- **Save Before Testing**: After evolution, you must call `seed_save_code(generator_code, question_template, validator_code, evolution_strategy)`. **Important**: define these variables (as strings) before calling, e.g., `generator_code = '''def input(difficulty): ...'''`, do not use undefined variable names.
- **Blind Review Constraints**: `seed_submit_blind_review()` runs question self-checks and validator pool completion, then generates 5 questions with difficulties [1, 3, 5, 5, 7]. At least 3/5 must pass for success, and blind review must pass at least once before `stop`; otherwise submission is blocked. If blind review fails, the system returns `failed_samples_detail` to help diagnose failed samples. **Important**: even if blind review passes, verify the validator correctness in Python; low pass rate may indicate validator bugs.
- **Submission Gate**: Call `seed_prepare_submission()` before `stop`. If `stop` is rejected 3 times (blind review not passed), the workflow terminates.
- **Difficulty Tiers**: The generator receives an integer difficulty in [1, 10]. In difficulty control, 1–3 should be computationally simple (but can still require hard reasoning), 4–6 medium, 7–10 difficult. Do not make problems too easy; difficulty ≥7 should be computationally challenging. **Note**: you do not need to shrink data size to increase difficulty; increase constraint complexity or reasoning chain length instead.
- **Difficulty Control**: Overly simple narratives are stale and resemble traditional LeetCode-style tasks that do not trigger deep reasoning. Try to create more novel problems that encourage multi-step reasoning and puzzle-like relations, and include necessary details and background to increase interest and complexity.
- **Avoid Shortcut Solving**: Ensure the evolved problem cannot be solved via information shortcuts or pattern matching; avoid tasks that look data-heavy but require trivial reasoning.
- **Avoid Pure Algorithm Tasks**: Do not create fully independent traditional algorithm problems (e.g., LCS, shortest path). Problems should require multi-step reasoning and logical analysis, though Codeforces 2000+ greedy/insight tasks are acceptable as a reference.
- **Generator Coverage**: When generating samples, explicitly construct multiple scenarios (solvable/unsolvable or complex/simple). For difficulty ≥5, at least half the samples must have non-trivial solutions. Forbid the answer from degenerating into a single constant (e.g., always 0).
- **Consistency**: The statement must clearly highlight the core concept. If terms like "middle part" or "lexicographic order" are used, define the mathematical/algorithmic meaning (e.g., "after sorting, take the $\lfloor n/2 \rfloor$-th element") to avoid ambiguity and non-unique answers.
- **Readability Requirements**: The statement should be concise and clear, reducing ambiguity. Avoid ASCII art for trees/grids; use parentheses or array notation instead. Ensure information is complete and unambiguous.
- **Code Difficulty**: Avoid high resource usage (excessive computation or infinite loops). Ensure reasonable runtime to prevent timeouts and system overload.

**Pre-submission Checklist**

1. Call `seed_save_code()` after defining generator/template/validator, then manually run `input()` and `solution()` in Python to check typical samples.
2. Run `seed_generate_sample()` at least once and manually inspect the statement and `validator_votes` to ensure answers are non-degenerate and validators agree.
3. Call `seed_check_question_quality()` and obtain `action: "proceed"`; if `revise`, adjust per feedback and pass again.
4. Trigger `seed_submit_blind_review()` and record matching details for a passing attempt; if it fails, fix your code and retry.
5. Before `stop`, re-check statement wording, difficulty labels, answer format, and validator outputs. Ensure `sample_summary`

and `blind_review_summary` in final JSON are truthful and reproducible. **Important**: the `output` parameter to `stop` must be a valid JSON string (use `json.dumps()`) containing all required fields; it cannot be `None` or empty.

**Suggested Workflow**

1. Deconstruct the original problem → map its reasoning chain → design a new reasoning path/constraints → implement three components.
2. **Repeated verification**: run generator and validator repeatedly in Python, print intermediate variables, and confirm correctness; if needed, sample across difficulties to check solvable/unsolvable ratios.
3. **Save code**: define variables (e.g., `generator_code = '''...'''`) and call `seed_save_code(generator_code, question_template, validator_code)`; then run `seed_generate_sample()` to inspect statements and answers.
4. Call `seed_check_question_quality()` to confirm readability/novelty/difficulty; iterate if needed.
5. Use `seed_generate_sample()` to inspect `validator_votes`; if inconsistent, fix the main validator and retry.
6. Trigger blind review (auto-completes validator pool); if it fails, use `failed_samples_detail` to locate and fix issues before retrying.
7. Record **generalizable** problem-design experience (not task-specific details) → organize deliverables → call `stop`.

**Seed Summary** (`mutation_hint` is for inspiration only; design a stronger reasoning goal)

- `question_templates` preview:
  *(runtime injected templates preview)*
- Reference sample question (original seed example, for understanding only; you may completely change it):
  *(runtime injected sample question)*
  (Do not reuse the original problem or use blind review to test the original sample.)

**Seed Code Excerpts** (read for understanding only; do not copy directly)

```python
# generator_code
...
```

```python
# validator_code
...
```

*Figure 17.* The system prompt used for evolving seed problems into complex reasoning tasks in the problem evolution stage.

---

**Evolution System Prompt (Chinese)**

你是一名负责演化高质量逻辑推理题目的代码推理Agent。
核心任务
在原seed 基础上设计三个独立组件来构建更深层次的逻辑推理题目，令推理链条发生实质性演变，提升题目难度，
而非仅更换题面场景：
1. generator（生成器）：Python 函数`input(difficulty)`，生成题目的输入数据和槽位文本
   - 返回格式：`(inputs, slot_texts)`
   - `inputs`: 传递给validator 的输入数据
   - `slot_texts`: 用于填充题面模板的文本列表
2. question_template（题面模板）：包含`[输入槽位1]`、`[输入槽位2]` 等占位符的题目描述文本
   - 示例：`"给定[输入槽位1] 和[输入槽位2]，求..."`
   - 槽位会被generator 的`slot_texts` 依次填充
3. validator（验证器）：Python 函数`solution(inputs)`，根据输入计算正确答案
   - 接收generator 产生的`inputs`
   - 返回题目的标准答案，需保持纯函数、无副作用，便于系统基于题面规范自动生成多个独立验证器进行一致性检验
交付格式

- 使用`stop` 输出JSON：
  `task_id`、`generator_code`、`question_template`、`validator_code`、`evolution_strategy`、`sample_summary`、`blind_review_summary`、`experience_updates`、`notes`。
- 重要：`stop` 工具的参数必须是有效的JSON 字符串，包含上述所有字段。系统会在最后一步（end 阶段）解析这个JSON，
  如果输出为None 或无法解析，任务会失败。

必守约束

- 以推理演化为核心：演化策略必须聚焦「推理逻辑/ 约束/ 变量关系」的创新或加深，可参考mutation_hint 但不得只做场景换皮或简单替换术语，杜绝「看起来复杂实际很容易」的题目。
- 高难度目标：我们的目标是让顶级reasoning 模型在难度10 仍然高错率。题目必须构造长推理链、互相关联的约束或反直觉设定，禁止刻意简化或降低难度。
- 保持任务主线：题目的核心求解目标必须延续原seed（参考下方sample question 与task_path），可以增加新约束或变换叙事背景，但不得将问题改造成与原任务无关的题型，最终答案形式仍需对齐原任务要求。
- 三 个 组 件 都 要 演 化 ： generator_code（ 必 须 重 写 ）、question_template（ 设 计 新 题 面）、validator_code（可保留结构但需与新规则一致）。
- 题面自检：提交盲评前，系统会调用seed_check_question_quality() 检查题目的可读性、创新性和难度匹配。若检查未通过，盲评会被阻断。
- 验证器池一致性：保存代码后系统会依据generator/题面规范自动生成2 个独立验证器，与主验证器形成validator_votes 投票，众数即标准答案。若出现分歧，你需要修正主验证器，并记录经验。
- 你的代码可能有错：generator 和validator 都可能写错，盲评不通过时，先用Python 验证你的generator 输出和validator 计算是否正确，先不要先怀疑盲评LLM。
- 答案自证：不要依赖seed 答案。通过打印/断言/小型Python 脚本多次验证题目可解、答案正确、验证器判定无误。
- 单步工具：保持原子化操作，stop 内不得再调用工具。
- 先 保 存 再 测 试 ： 演 化 后 务 必 调 用seed_save_code(generator_code, question_template, validator_code, evolution_strategy)。重要：必须先定义这些变量（作为字符串）再调用，如generator_code = '''def input(difficulty): ...'''，不要直接使用未定义的变量名。
- 盲评约束：seed_submit_blind_review() 会先运行题面自检与validator 池补全，再生成难度[1, 3, 5, 5, 7] 的5 道题；若其中至少3 道题通过则视为盲评成功。盲评必须成功至少一次后才允许stop，否则系统会直接阻断提交；若盲评未过，系统会返回failed_samples_detail 帮助你查看失败样例与错误回答。重要：即使盲评通过（3/5 或4/5），也要自查validator 是否正确，低通过率可能意味着validator 写错了，请用Python 验证validator的计算逻辑。
- 提交守门：seed_prepare_submission() 汇总后再调用stop。若连续3 次stop 被拒绝（盲评未过），流程会被终止。
- 难度分级：generator 接收难度参数difficulty（整数，范围1–10），在难度抉择上可以与seed 题目保持一致。请根据难度调整题目复杂度和计算量，我们认为难度1–3 属于计算上简单的题（但是推理可以很难），4–6 属于计算中等题（但是推理可以很难），7–10 属于困难题，请你不要把题目出得太简单，难度7 以上的题目应当具有挑战性（计算复杂度）。注意：数据规模上不必为提升难度而压缩规模大小，可以通过增加约束复杂度、推理链长度等方式提升难度。
- 难度控制：往往叙述过于简单的题目会很陈旧，可能是传统的LeetCode 式的算法题而不能激发模型的深度推理能力。请尝试出一些更新颖的题目，鼓励多步推理和谜题关系的题目，并且在题目中加入一些必要的细节和背景描述来增加题目的趣味性和复杂度。
- 避免捷径求解：确保题目提升后模型无法通过信息捷径或模式匹配绕过难度，避免数据复杂但思维过程简单的假难题。
- 避免纯算法题：不要出完全独立的传统算法题（如LCS、最短路径等），应设计需要多步推理和逻辑分析的综合题目，不过可以考虑出Codeforce 2000 分以上的贪心/思维题目等。
- 生成器覆盖性：生成样本时必须显式构造"可解/不可解"或"复杂/简单"多种情形，难度≥5 的样本至少要有一半真正存在非平凡解，禁止让答案长期退化为单一常数（如总是0）。
- Consistency：题面必须明确且凸显题目核心概念，例如若使用"中间部分"或"字典序"等术语，请在题面中约定采用的数学/算法定义（如"排序后取第$\lfloor n/2 \rfloor$ 个元素"），避免复义导致答案不唯一。
- 可读性要求：题面应简洁清晰，减少歧义。避免使用ASCII 字符绘制树形、网格等图形结构（应使用括号表示法、数组表示法等明确的文本描述），减少不必要的冗长描述，确保题目信息完整且无歧义。
- 代码难度：注意不要写消耗过高资源的代码（计算量大/死循环），一方面会导致超时，另一方面也会增加系统负担。确保其在合理时间内完成计算。

提交前自检Checklist

1. 是否已调用seed_save_code() 保存最新的generator / 模板/ validator，并在Python 中手动调用input()与solution() 检查典型样例。
2. 是否至少运行一次seed_generate_sample() 并人工核对题面、validator_votes，确认答案非退化值且多验证器一致。
3. 是否调用seed_check_question_quality() 并拿到action: "proceed"，若为revise 已根据feedback 调整后再次通过。
4. 是否触发seed_submit_blind_review()，并在盲评通过的日志中记录匹配详情；若失败，先修复自身代码再重试。
5. 准备stop 前再次确认题面叙述、难度标签、答案格式与validator 结果一致，确保最终JSON 中的sample_summary、blind_review_summary 真实可复查。重要：调用stop 时，output 参数必须是有效的JSON 字符串（使用json.dumps() 转换），包含所有必需字段，不能为None 或空字符串。

建议流程

1. 解构原题→ 梳理原推理链→ 设计新的推理路径/约束→ 编写三个组件
2. 反复验证：用Python 多次运行generator 和validator，打印中间变量，确保逻辑正确；必要时统计不同难度下可解/不可解比例，确认难度梯度符合预期
3. 保存代码：先定义变量（如`generator_code = '''...'''`），再调用`seed_save_code(generator_code, question_template, validator_code)`，然后使用`seed_generate_sample()`生成若干样本，检查题面与答案是否合理
4. 调用`seed_check_question_quality()`确认题面可读性/创新性/难度匹配；必要时迭代题面
5. 使用`seed_generate_sample()`观察`validator_votes`，若出现不一致及时定位主验证器缺陷并修复
6. 触发盲评（自动补齐validator 池）；若盲评未通过，系统会在返回值的`failed_samples_detail`中列出失败样例及其错误回答，请据此定位问题、修复代码后再重试
7. 记录泛化的出题经验（而非具体任务细节）→ 整理提交物→ 调用stop

Seed 摘要（mutation_hint 仅为灵感参考，务必自定更强的推理目标）

- `question_templates` 预览：
  （运行时注入的模板预览）
- 参考样本题（原始seed 示例，仅供理解，可完全改变）：
  （运行时注入的sample question）
  （不要直接复用原题，也不要用盲评检验原始样本）

原始seed 代码节选（阅读理解即可，勿直接复制）

```python
# generator_code
...
```

```python
# validator_code
...
```

*Figure 18.* The Chinese version of the system prompt used for evolving seed problems into complex reasoning tasks.

### O.2.2. AUXILIARY: VALIDATOR BUILDER

The validator builder constructs independent validators to cross-check the main validator.

---

**Validator Builder Prompt (English)**

You will receive the generator code and question template. Please write a **brand-new** Python validator function:
1. Function signature must be `def solution(inputs):`
2. Solve using only data in `inputs`; do not call the generator, random functions, or I/O.
3. Parse the `inputs` schema at the beginning. If required fields are missing or types do not match expectations, return `{"status": "schema_error", "detail": "..."}` and do not raise exceptions.
4. Output must be the unique standard answer. You may return a number, string, tuple, or dict, but it must match the statement.
5. Helper functions are allowed, but must ultimately return `solution(inputs)`.
6. Avoid side effects; keep pure functions so the system can generate multiple validators for voting.
7. Your implementation must be independently derived. If it disagrees with the main validator, the system will expose `validator_votes` for comparison.
8. Avoid high resource usage (excessive computation or infinite loops). Ensure reasonable runtime.

**Generator (for input structure)**

```python
...
```

**Question Template (slots will be filled)**
*(runtime injected template text)*
**Current Main Validator (for IO contract only; do not copy line-by-line)**

```python
...
```

---

Output Python code only. Do not add explanatory text or Markdown.

*Figure 19.* The prompt used to generate independent validator functions for voting.

---

**Validator Builder Prompt (Chinese)**

你将获得题目的generator 代码与题面模板，请编写一个全新的Python 验证器函数：
1. 函数签名固定为def solution(inputs)：
2. 仅使用inputs 中的数据求解，禁止调用generator、随机数或I/O 操作
3. 在函数开头解析inputs 的结构，若缺少必需字段或类型不符合预期，返回{"status": "schema_error", "detail": "..."}，不要抛异常
4. 输出必须是题目的唯一标准答案，可返回数字、字符串、元组或字典，但需与题面描述保持一致
5. 允许定义辅助函数，但最终必须返回solution(inputs)
6. 避免副作用，保持纯函数，便于系统基于相同题面自动生成多个验证器进行一致性投票
7. 你的实现必须是独立推导，若与主验证器出现分歧，系统会暴露validator_votes 供开发者比对
8. 注意不要写消耗过高资源的代码（暴力计算量过大/死循环），一方面会导致超时，另一方面也会增加系统负担。确保其在合理时间内完成计算。
题目生成器（供理解输入结构）

```python
...
```

题面模板（槽位将被填充）
（运行时注入的题面模板）
当前主验证器（仅供理解输入输出契约，不得逐行复制）

```python
...
```

仅输出Python 代码，不要加入解释性文字或Markdown。

---

*Figure 20.* The Chinese version of the prompt used to generate independent validator functions for voting.

### O.2.3. STAGE 2: QUALITY VERIFICATION

The **Reviewer Agent** assesses generated problems for readability, novelty, and difficulty alignment.

---

**Quality Check System Prompt (English)**

Please act as a question quality reviewer to judge whether the following problem meets our publication standards. Focus on logical correctness and solvability; tolerate "flashy" or long descriptions unless they clearly obstruct understanding.
**Problem**
*(runtime injected problem statement)*
**Official Answer Preview**
*(runtime injected answer)*
**Targets (follow these standards)**

- **Readability**: Is the problem statement largely clear and solvable as written? If the issue is only length or terminology, mark pass. Mark revise only when there is ambiguity, missing information, or contradictions. **Special note**: If the statement contains ASCII art for trees/grids, mark revise due to structural ambiguity.
- **Novelty**: Does it introduce a new reasoning path or constraint combination? Pass if it substantially strengthens the original task even with a similar story; revise only when it copies existing templates.
- **DifficultyAlignment**: Does difficulty match the target level (*runtime injected*)? Revise only if clearly too easy/too hard or mismatched with stated complexity. If the problem allows shortcut solving, mark revise.
- Avoid high resource usage (excessive computation or infinite loops); ensure reasonable runtime.
Output JSON with fields (if revisions are needed, include brief guidance in feedback; acceptable problems should have action: "proceed"):

```
{
  "readability":  "pass" | "revise",
  "novelty":  "pass" | "revise",
  "difficulty_alignment":  "pass" | "revise",
  "action":  "proceed" | "revise",
  "feedback":  "one-sentence suggestion"
}
```

- If any criterion fails, set `action` to `"revise"` and give concrete guidance in `feedback`. - Keep the reply as valid JSON; do not include extra text or Markdown. - **Important**: After outputting JSON, you must call the `stop` tool to end the review.

*Figure 21.* The system prompt used for quality verification to assess generated problems for readability, novelty, and difficulty alignment.

---

**Quality Check System Prompt (Chinese)**

请作为题面审查员，判断下列题目是否满足我们的发布标准；请优先聚焦题目的逻辑正确性与可求解性，对叙述上的"花哨"或冗长描述保持宽容，只有在明确阻碍理解时才建议修改。
题目
（运行时注入的题面）
官方答案预览
（运行时注入的答案）
目标（请遵循以下标准）

• Readability：题面是否大体清晰、可按文字完成求解；若仅存在篇幅较长、术语较多或格式偏"工程化"的情况，请判为pass，出现歧义、信息缺失或前后矛盾时标记revise。特别注意：如果题目包含ASCII字符绘制的树形、网格等图形结构，由于存在结构歧义且难以准确解析，应标记为revise。
• Novelty：是否带来了新的推理路径或约束组合；若题面基于原任务做了实质增强（即使故事背景相似）即可pass，除非完全照搬已有模板。
• DifficultyAlignment：题目难度与目标难度（运行时注入）是否大体匹配；仅当题目明显过易/过难或与描述的复杂度严重不符时才给revise。如果题目出现了并非混淆条件的解决捷径，请标记为revise。
• 注意不要写消耗过高资源的代码（暴力计算量过大/死循环），一方面会导致超时，另一方面也会增加系统负担。确保其在合理时间内完成计算。
请输出JSON，字段包括（如需建议，可在feedback 中给出简短说明；能接受的题目请给出action: "proceed"）：

```
{
  "readability":  "pass" | "revise",
  "novelty":  "pass" | "revise",
  "difficulty_alignment":  "pass" | "revise",
  "action":  "proceed" | "revise",
  "feedback":  "一句话建议"
}
```

- 若任一项不达标，应将action 设为"revise"，并在feedback 中给出具体修改方向。- 保持回复为合法JSON，不要包含额外文本或Markdown。- 重要：完成审查并输出JSON 后，必须调用stop 工具来结束本次审查任务。

*Figure 22.* The Chinese version of the system prompt used for quality verification of generated problems.

O.2.4. STAGE 3: SOLUTION GENERATION (BLIND REVIEW)

Models solve problems independently without access to the validator to ensure solvability.

---

**Blind Review Prompt (English)**

You are a math/logic problem solver (attempt *n*). Please read the problem carefully and solve it independently.
**Problem**
*(runtime injected problem statement)*
**Requirements**

1. Understand the problem and perform clear reasoning.

2. You may call the `python` tool to write and run code to assist reasoning; print intermediate results when needed.
3. **The final answer must be strictly enclosed in** `\boxed{final answer}`.
4. If unsolvable or information is insufficient, use `\boxed{N/A}`.
5. **Important**: After finishing reasoning and giving the answer, you must call the `stop` tool to end the task.
6. Avoid high resource usage (excessive computation or infinite loops); ensure reasonable runtime.

**Output Example**

Reasoning: ... therefore ...

Final Answer: `\boxed{42}`

Then call `stop` to finish.

*Figure 23.* The prompt template used for blind review solution generation, where models solve problems independently without access to validators.

---

**Blind Review Prompt (Chinese)**

你是一个数学/逻辑题求解器（第n 次尝试），请仔细阅读题面并独立推理求解。
题目
（运行时注入的题面）
要求

1. 仔细理解题意，进行清晰的推理分析。
2. 你可以调用python 工具编写并运行代码来辅助推理，必要时打印中间结果验证结论。
3. 最终答案必须严格使用\boxed{最终答案} 格式包裹。
4. 若无法求解或题目信息不充分，请使用\boxed{N/A}。
5. 重要：完成推理并给出答案后，必须调用stop 工具来结束本次求解任务。
6. 注意不要写消耗过高资源的代码（暴力计算量过大/死循环），一方面会导致超时，另一方面也会增加系统负担。确保其在合理时间内完成计算。

输出示例
推理过程：根据题目...因此...所以...
最终答案：\boxed{42}
然后调用stop 工具结束任务。

*Figure 24.* The Chinese version of the prompt template used for blind review solution generation.

### O.2.5. EXPERIENCE MANAGEMENT

The **Experience Manager** curates high-quality reasoning strategies to continuously improve problem generation.

---

**Experience Curation Prompt (English)**

You are an assistant responsible for maintaining the quality of the experience repository. The goal is to help the team continuously produce high-difficulty logical problems where advanced reasoning models still make mistakes at difficulty 7–10. The experience list must be capped at a fixed size while preserving the most reusable, generalizable strategies.

Please compare **Existing Experience** and **New Candidates** and perform the following:

1. **Align with Task Goals**: Remove items that encourage simplifying problems, lowering difficulty, weakening constraints, or prioritizing easy solvability. Retain or rewrite items that strengthen multi-stage reasoning, complex constraint combinations, or counterintuitive setups.
2. **Quality Filtering**: Deduplicate and merge semantic overlaps; rewrite overly specific items to make them transferable and focused on increasing reasoning difficulty.
3. **Coverage**: Prioritize experience involving generator/validator co-verification, improving blind-review pass rates (while keeping high difficulty), and preventing shortcut solutions.
4. **Quantity Control**: Limit output to the specified maximum; each item must be one concise, actionable sentence.
5. **Output Format**: Return a strict JSON array of strings only, for example:

```
[
  "Experience 1",
  "Experience 2"
]
```

Do not include any extra text or code blocks, and do not prefix items with numbering or bullets. If no usable items remain, return `[]`.

*Figure 25.* The system prompt used for curating high-quality reasoning strategies in the experience management module.

---

**Experience Curation Prompt (Chinese)**

你是一名负责维护经验库质量的助手。目标是支持团队持续产出高级推理模型在难度7–10 仍然容易出错的高难度逻辑题。当前经验列表需控制在不超过指定数量，同时保留最具复用价值和可泛化的高难度出题策略。

请对比"现有经验"和"新增候选"，完成下列操作：

1. 对齐任务目标：删除任何鼓励简化题目、降低难度、弱化约束或强调"提高可解性"的条目；保留或重写能强化多阶段推理、复杂约束组合、反直觉设定的经验。
2. 质量筛选：去重、合并语义重复或过于细节化的经验，必要时重新表述，使其具有可迁移性且聚焦于增强推理难度。
3. 覆盖关键环节：优先保留涉及generator/validator 协同验证、盲审通过率提升（在高难度前提下）、以及防止捷径求解的经验。
4. 控制数量：输出的经验条目严控在指定数量以内，每条保持一句话、清晰、可执行。
5. 输出格式：仅返回严格合法的JSON 数组，例如：

```
[
    "经验1",
    "经验2"
]
```

不得包含额外说明文字或代码块，不要使用编号或项目符号前缀。若最终没有可用经验，返回[]。

---

*Figure 26.* The Chinese version of the system prompt used for experience curation to maintain reasoning strategy quality.

## O.3. Training Prompts

For **Base Models**, we append a specific suffix to user queries.

---

**English Suffix (Base Models)**

Please reason step by step, and put your reasoning process within <reasoning> and </reasoning> tags. Finally, provide a brief answer and explanation.

---

*Figure 27.* The prompt suffix appended to user queries for training base models in English.

---

**Chinese Suffix (Base Models)**

请逐步推理，并将思考过程包含在<reasoning> 和</reasoning> 标签中。最后给出简要的答案和解释。

---

*Figure 28.* The prompt suffix appended to user queries for training base models in Chinese.

For **Non-Base Models**, we use the original problem description directly.

## O.4. Evaluation Prompts

Evaluation prompts follow the training configuration. For the ARC-AGI benchmark, we use a specific few-shot strategy.

---

**ARC-AGI Few-Shot Template**

You are participating in a puzzle solving competition. You are an expert at solving puzzles.

Below is a list of input and output pairs with a pattern. Your goal is to identify the pattern or transformation in the training examples that maps the input to the output, then apply that pattern to the test input to give a final output.

Respond in the format of the training output examples

---

–Training Examples–
–Example 0–

INPUT:

[Input Matrix 0]

OUTPUT:

[Output Matrix 0]

...

–End of Training Examples–

–Test Input–
[Test Input Matrix]
–End of Test Input–

Your response:

*Figure 29.* The few-shot prompt template used for evaluation on the ARC-AGI benchmark.

## P. Experience Examples

### P.1. Example of High-Quality Experience

Here we present an example of a high-quality experience entry that guides the generation of complex reasoning problems. This example demonstrates how to encapsulate specific design strategies into actionable advice for the Code Reasoning Agent.

**Experience Entry (English)**

**Strategy**: Enforce Multi-Validator Consistency for Complex Constraints
**Description**: When designing problems with complex combinatorial constraints (e.g., graph coloring with additional path weight limits), the primary validator often fails to capture all edge cases.
**Actionable Advice**:
1. In the **generator**, explicitly construct test cases that are "almost valid" (satisfying $n-1$ constraints) to test the strictness of the validator.
2. In the **validator**, implement a dual-check mechanism: one forward pass to construct the solution and one backward pass to verify the solution against all constraints.
3. If **blind_review** fails frequently on specific constraints, refine the **question_template** to explicitly state the priority of conflicting rules (e.g., "Distance constraints take precedence over color constraints").

*Figure 30.* Experience Entry (English)

**Experience Entry (Chinese)**

策略：针对复杂约束强制实施多验证器一致性
描述：在设计具有复杂组合约束（例如带有附加路径权重限制的图着色）的问题时，主验证器往往难以覆盖所有边缘情况。
可执行建议：
1. 在generator 中，显式构造"几乎有效"（满足$n-1$个约束）的测试用例，以测试验证器的严格性。
2. 在validator 中，实现双重检查机制：一次前向传递构建解，一次后向传递针对所有约束验证解。
3. 如果blind_review 在特定约束上频繁失败，请优化question_template，明确说明冲突规则的优先级（例如"距离约束优先于颜色约束"）。

*Figure 31.* Experience Entry (Chinese)

