# OpenReview forum: "Scaling the Scaling Logic: Agentic Meta-Synthesis of Logic Reasoning"
_ICML.cc/2026/Conference — ICML 2026 regular_

### Official Review · Reviewer_pWuh · 2026-03-06

**Soundness:** 2
**Presentation:** 1
**Significance:** 2
**Originality:** 3
**Overall Recommendation:** 4
**Confidence:** 3

**Summary:**

The paper introduces SSLogic, an agentic meta-synthesis framework designed to scale training data for Reinforcement Learning. The framework aims to evolve logical task families programmatically rather than relying purely on manual curation or fixed templates. Operating through a Generate-Validate-Refine loop, the pipeline expands an initial set of 400 human-annotated seed families into 953 families. To filter out low-quality or unsolvable data, the authors propose a Multi-Gate Validation Protocol that includes static quality checks, consensus-based dynamic evaluation via multiple generated solvers, and code-augmented blind review. The authors evaluate their synthesized dataset by training Qwen3-8B models via GRPO, reporting performance changes across several logic and mathematics benchmarks compared to models trained solely on the seed data.

**Compliance With Llm Reviewing Policy:**

Affirmed.

**Final Justification:**

The authors engaged in a constructive discussion and addressed most of my concerns about the paper.

The authors addressed the critical concern regarding the Thinking model's performance degradation when training on Seed. I have remaining concerns about whether longer training on the Evolve+Thinking would also improve, but it would be surprising if not. (The authors unfortunately did not provide this experiment; I assume they overlooked my question for it). The authors should add the results on extended training on thinking (all benchmarks, Seed **and** Evolve) to the final version of the paper.

The second major concern over human-annotation bottlenecks was addressed by using Enigmata as Seed data (reviewer 5GuQ). The scaling factor still seems somewhat limited, especially given the authors' promise of double scaling in their title. But more importantly, they convinced me that the evolved data is more difficult and of higher quality to train on.

The survivorship bias introduced by the Multi-Gate Validation Protocol remains an acknowledged limitation; however, if this is addressed in the paper, I don't see it as a crucial weakness.

All presentation issues have been addressed by the authors, and they have promised to address them in the final version.

**Key Questions For Authors:**

Please address the highlighted questions in the weaknesses.

**Limitations:**

yes

**Strengths And Weaknesses:**

Strengths

- The paper targets a critical problem for the machine learning community: the data bottleneck for reasoning tasks. Shifting the generation target from instance-level parameter perturbations to task-family-level evolution is a conceptually ambitious and valuable research direction.

- The proposed Multi-Gate Validation Protocol demonstrates a sophisticated engineering effort. By incorporating consensus checks and an adversarial blind review, the pipeline makes an empirical attempt to mitigate the hallucination risks and logical inconsistencies typical of LLM-driven data generation.



Weaknesses

1. Conceptual and Methodological Limitations

    - The pipeline relies heavily on an initial foundation of 400 "human-annotated" seed families. Please document the origin of these 400 seeds and the human engineering cost to create these seed families. Given that the approach creates only 1.38 families per seed family, the framework's scale remains heavily dependent on the size of the manually authored seed pool. Furthermore, the reported amortized cost of $1.18 per accepted task family appears to account solely for API and compute usage, omitting the human labor required to author the seeds. **Q1:** To substantiate the claim that this framework resolves the manual data bottleneck, the authors must transparently detail the human annotation costs and clarify whether the method truly scales independently or simply leverages an initial human investment.

    - In Section 5.4, the authors note that 50% of rejected tasks fail due to "Implementation Bugs". While the Multi-Gate Validation Protocol successfully filters these out, the authors don't address the conceptual limitation this imposes on the framework's scope. Specifically, synthesizing a programmatic solver is usually more difficult than solving a single instance of a puzzle, introducing a survivorship bias for less complex puzzles. **Q2:** The authors should explicitly discuss this as a conceptual limitation of their paradigm.

    - The authors use prompt-based instructions and an LLM Reviewer Agent to ensure that the newly generated 'Evolve' tasks are structurally novel and not only surface-level variations of the original Seed tasks. The methodology would be significantly strengthened by including overlap metrics between the Seed and Evolve datasets. **Q3:** Reporting these metrics would provide concrete evidence of the pipeline's ability to generate genuinely novel logic structures.

2. Experimental Limitations and Flaws

    - The authors claim the negative control experiment using ARC-AGI proves that tasks lacking explicit intermediate reasoning chains (input-output mappings) interfere with rigorous mathematical deduction. This conclusion is scientifically unsupported. ARC-AGI is in a fundamentally different domain (2D spatial/visual grids) compared to AIME. The total performance degradation is more likely attributed to the domain switch than a result of missing reasoning steps. In fact, the training on ARC-AGI does improve results on the ARC-AGI benchmark (and at least does not degrade on the Logic Benchmark total), which actually invalidates the hypothesis. The section rather reads as a failed experiment framed as control study.

    - The authors explore the effectiveness of their training dataset on Qwen3-8B(Thinking) to explore performance ceilings. The performance here is underwhelming. the training on Seed is actively hurting the models performance (at least on the logic benchmark, although the Seed families are of logical nature), while Evolve barely improves the no-finetuning baseline. I believe that is is an indication that their approach is not compatible with fine-tuning thinking models. **Q4:** The authors need to address this more explicitly. It also lessens the approaches significance as all models that perform adequately on such reasoning tasks are "thinking/reasoning models".

    - The conclusions drawn in Section 5.3 regarding why the framework drives reasoning improvement are overstated and lack direct empirical support. The authors use code complexity metrics of the synthesized Python scripts as a proxy for the intrinsic logical difficulty of the generated puzzles. However, programmatic complexity does not reliably correlate with cognitive or logical complexity. Consequently, the assertions that the dataset "transcends human-authored seeds by fundamentally shifting the structural density and algorithmic breadth of the training signal" or "forces explicit System 2 behaviors" are speculative. The authors must temper such claims and acknowledge the limitations of using code syntax metrics as a proxy for reasoning depth.


3. Presentation

    - The paper overstates the rigor of its training signals by labeling them "verifiable." The validators/solvers are synthesized by LLM, without formal or factual guarantees. To avoid misleading readers, the authors should clarify this limitation and use more accurate terminology.

    - The presentation is overly labored; the authors frequently substitute descriptive language for bloated, mathematically hand-wavy, and partly invented terminology (e.g., "composite gating," "canonical solver", "dynamic verifiability") and pseudo-mathematical abstractions. The paper would be significantly improved by adopting clearer, more direct descriptive language to explain the pipeline's mechanics.

    - The Appendix documents prompts in both English and Chinese. However, the authors do not adress this in the main paper. It is unclear whether the Seed families, the generated Evolve tasks, and the prompts were conducted in English, Chinese, or a mixture of both. **Q5:** The authors must explicitly document the language(s) used throughout the pipeline. Furthermore, the authors should clarify which of the prompts provided in the appendix are the originals or translations. If the pipeline operates multilingually or if language choice significantly impacts the generation success rate, the authors should discuss this and ideally provide an ablation.

    - **Q6:** Is there a difference between Evolve and SSLogic-Evolve? It seems, the authors inconsistently refer to its generated dataset as both "Evolve" and "SSLogic-Evolve". Alternating between them introduces unnecessary confusion, potentially leading readers to believe these are distinct datasets or ablations. The authors should standardize this nomenclature throughout the manuscript for clarity.

---

> ### Author Rebuttal · Authors · 2026-03-31
>
> We thank the reviewer for recognizing the importance of the data bottleneck problem and the sophistication of the Multi-Gate Validation Protocol.
>
> **[Q1] Human annotation cost and scalability.**
>
> The 400 seed families were created by full-time annotators (all holding bachelor's degrees) as part of a broader research effort. Since these annotators work on salary (not per-task payment), the annotation cost is not meaningfully separable as a standalone line item — similar to how AgenticMath(MATH) [1] and other data synthesis works(e.g. generative augmentation on ImageNet) do not attribute per-item human costs for their seed data. The **$1.18 per family reports the marginal cost** of evolution, which is the relevant metric for scaling. More importantly, the framework is **seed-agnostic**: our Enigmata experiment (detailed in our response to Reviewer 5GuQ, W1/Q1) demonstrates comparable improvements on an open-source dataset **without any human annotation**.
>
> **[Q2] Survivorship bias.**
>
> Our data shows that rejected families are actually **harder**, not simpler:
>
>  Model | Tasks | avg@8
> ---|---|---
>  DeepSeek-v3 | 148 | 0.477
>  Doubao-Seed-1.6-Thinking | 148 | 0.556
>
> Both scores are lower than the accepted pool average. The pipeline's coverage ceiling is bounded by the synthesis model's code-generation capability, not by framework design. As models improve, coverage naturally extends to harder families.
>
> **[Q3] Overlap metrics.**
>
> Rather than surface-level code similarity metrics — which conflate code style with logical substance — we demonstrate novelty through **category-selective downstream impact**. On KORBench and BBH (new analysis, detailed in our response to Reviewer 5GuQ, Q3): Logic +13.2%, Operation +9.6%, Cipher +9.2%, but Puzzle +1.2%. Surface-level paraphrasing would produce uniform gains; the selective pattern indicates genuinely novel reasoning structures. Section 5.2 (Figures 6-7) corroborates: Evolve shows 2-3x higher cyclomatic/cognitive complexity and substantially higher DP, graph, and sorting pattern rates.
>
> **[W4] ARC-AGI negative control.**
>
> We included this experiment to follow Enigmata [2], which explored combining ARC-AGI with logic reasoning for RL training. We beg to differ with the characterization that our results "invalidate the hypothesis." The reviewer notes that ARC-AGI training improves ARC-AGI results — this is precisely our point. ARC-AGI training helps ARC-AGI but does not transfer to logic reasoning, demonstrating that **not all verifiable tasks are interchangeable as RL training signal**. The domain mismatch is not a confound; it is the finding. Recent work confirms ARC-AGI is fundamentally a vision task [3]. We will sharpen the framing.
>
> **[Q4] Thinking model compatibility.**
>
> The characterization "Seed is actively hurting performance" reflects only a subset of benchmarks. Table 2C shows Seed@240 gains on math: AIME24 +2.6, Brumo25 +2.7, HMMT25 +1.9. Evolve@240 shows continuous improvement throughout training, with Macro Accuracy rising from 0.4674 to **0.5638 (+20.6%)** ([Figure R5](https://anonymous.4open.science/r/anonymousPDF-0257/R5_Q4_thinking_training_dynamics.png)). For Seed, extended training at 3.33x compute confirms full compatibility:
>
>  Benchmark | No-train | Seed\@3.3x | Delta
> ---|---|---|---
>  AIME24 | 71.7 | 75.7 | +4.0
>  AIME25 | 63.2 | 69.1 | +5.9
>  HMMT | 40.9 | 43.3 | +2.4
>  BBH | 89.3 | 90.2 | +0.9
>  BBEH | 30.0 | 31.1 | +1.1
>
> Thinking models require more compute for policy adaptation — an expected finding given their strong starting point, not a fundamental incompatibility.
>
> **[W6] Section 5.3 overclaiming.**
>
> Section 5.2 presents code metrics as structural descriptors. We accept that certain phrases in 5.3 overstate the evidence and will revise to correlational framing. The KORBench category-selective gain pattern (Q3) provides stronger correlational evidence than code metrics alone.
>
> **[Q5] Language documentation.**
>
> Seed families are in Chinese; evolved tasks inherit the language. Appendix provides both — Chinese is original, English is translated. All evaluations use English benchmarks, and improvements transfer across languages. Our manual audit of agent traces found no synthesis failures attributable to language inconsistency (see Reviewer 5GuQ, Q2 for audit details). Our Enigmata experiment further confirms the pipeline works equally on English-only data.
>
> **[Q6] Naming.**
>
> "SSLogic-Evolve" is abbreviated as "Evolve" in space-limited tables. Both refer to the same dataset. We will standardize.
>
> [1] Yue et al., "AgenticMath: Enhancing LLM Reasoning via Agentic-based Math Data Generation," 2025.
>
> [2] Chen et al., "Enigmata: Scaling Logical Reasoning in Large Language Models with Synthetic Verifiable Puzzles," NeurIPS 2025.
>
> [3] Hu et al., "ARC Is a Vision Problem!", 2025.

---

> > ### Author Rebuttal · Reviewer_pWuh · 2026-04-02
> >
> > The authors have addressed most of my concerns appropriately. Please elaborate on the remaining concerns:
> >
> > Q1: I understand that the human annotation cost cannot be documented. I still find it difficult to assess whether the method is truly enabling scaling, as the approach creates only 1.38 families per seed family.
> >
> > Q2: My question was to address the following: "Specifically, synthesizing a programmatic solver is usually more difficult than solving a single instance of a puzzle, introducing a survivorship bias for less complex puzzles." Can the authors please elaborate on their interpretation of the data? If rejected families are harder (i.e., remaining families easier), this is a strong indication of a survivorship bias. The programmatic solver (validator) is an integral part of the framework as a filter. If this filter indeed introduces a survivorship bias, it is a weakness in the framework's design.
> >
> > Q3: Thank you for the interesting additional experiments. I agree with the authors that this is a strong indication of genuinely novel logic structures in the evolve set.
> >
> > W4:Thanks for the explanation. Given the results, the ARC-AGI-Evolve does transfer to other logic tasks, still improving (slightly) on all except BBH. However, since these improvements are marginal and the mathematical benchmarks clearly degrade, I agree with the author's general direction.
> > Given the linked paper (ARC Is a Vision Problem!) I also understand the authors' speculative explanation in the paper. However, this explanation would need a more detailed discussion in the final paper. Empirical findings supporting this hypothesis would be optimal.
> >
> > Q4: Thanks for the experiment with longer training. This is indeed a positive outlook. I am interested in the corresponding Evolve@3.3x results, and, more importantly, the Seed@3.3x evaluations on the missing ARC-AGI and Enigmata - on those, the degradation on seed was most pronounced. For the final version, I would recommend replacing the Seed and Evolve @240 with the longer training.

---

> > > ### Author Response · Authors · 2026-04-03
> > >
> > > We thank the reviewer for the constructive follow-up. We address each remaining point below.
> > >
> > > **[Q1] Scalability beyond family ratio.**
> > >
> > > The 1.38x ratio measures family-level expansion (400→553) after two rounds, but each evolved family generates a fresh distribution of difficulty-controllable instances, yielding 3.7x instance-level expansion (5,718→21,389). The marginal cost ($1.18/family) amortizes over this larger pool, so the per-family ratio understates practical scaling.
> > >
> > > Moreover, the cold-start pool need not be human-authored. Our Enigmata experiment (Round 1, 5GuQ) shows comparable gains from a fully open-source seed with zero human annotation. Similar repositories exist (e.g., _Reasoning-Gym_ [1] with 100+ families). The scaling path is: aggregate open-source G-V pools → Evolve → iterate. The ratio is composable across rounds and seed sources, not a fixed ceiling.
> > >
> > > **[Q2] Survivorship bias.**
> > >
> > > The reviewer's reasoning is valid: the synthesis model's capability does impose a ceiling, and families beyond it are filtered out. We acknowledge this as a genuine coverage limitation.
> > >
> > > That said, we believe this does not invalidate training value. The key evidence is **where** gains concentrate. If the filter predominantly admitted easier families, we would expect uniform gains across categories, largest on already-easy tasks. Instead, gains are highly selective:
> > >
> > > - KORBench: Logic +13.2%, Operation +9.6%, but Puzzle +1.2%
> > > - BBH: multi-step tracking sub-tasks +44–56 pts, while near-ceiling sub-tasks (boolean_expressions 93.6→94.4) barely move
> > >
> > > This selectivity suggests surviving families introduce new _coverage_ in multi-step reasoning and algorithmic structures — dimensions Seed under-represents — rather than easier versions of existing tasks. The bias is better characterized as a coverage limitation (cannot yet reach certain hard families) than quality degradation of what it produces.
> > >
> > > Coverage ceilings are inherent in all automated synthesis paradigms: SynLogic is bounded by template authors' expertise; Enigmata by designer domain coverage. Our approach's ceiling rises automatically as code-generation models improve. We will discuss this in the Limitations section.
> > >
> > > **[Q3]** We thank the reviewer for the positive assessment.
> > >
> > > **[W4] ARC-AGI discussion.**
> > >
> > > We will expand the discussion in the revised version with extended training evidence (see Q4) and a more careful distinction between domain-transfer and reasoning-chain format effects.
> > >
> > > **[Q4] Seed@3.3× on ARC-AGI and Enigmata.**
> > >
> > > Complete Seed@3.3× results, including the two benchmarks the reviewer requested:
> > >
> > >  Benchmark    | No-train | Seed@3.3× | Delta
> > > --------------|----------|-----------|----------
> > >  AIME24       | 71.7     | 75.7      | +4.0
> > >  AIME25       | 63.2     | 69.1      | **+5.9**
> > >  HMMT25       | 40.9     | 43.3      | +2.4
> > >  BBH          | 89.3     | 90.2      | +0.9
> > >  BBEH         | 30.0     | 31.1      | +1.1
> > >  **Enigmata** | 39.4     | 41.2      | **+1.8**
> > >  **ARC-AGI**  | 7.6      | 3.8       | **−3.8**
> > >
> > > Enigmata recovers and surpasses the no-training baseline under extended compute (+1.8). For ARC-AGI, the continued degradation is consistent with our negative control analysis (Section 4.6): logic-focused RL optimizes for multi-step reasoning chains, competing with ARC-AGI's visual pattern-matching strategy. As the reviewer acknowledged in W4 (Round 1), this aligns with the finding that ARC-AGI is fundamentally a vision task.
> > >
> > > Evolve@3.3× is ongoing (~4,000 GPU-hours). Longer-training results are useful complementary evidence for clarifying sensitivity to training budget. We prefer to retain the fixed-step comparison as the ablation analysis and report longer-training results as a main check in the revised version.
> > >
> > > [1] Stojanovski et al., "Reasoning Gym: Reasoning Environments for RL with Verifiable Rewards," arXiv, 2025.
> > >
> > > ---
> > >
> > > **A note to the reviewer.**
> > >
> > > We sincerely appreciate the rigor of your review across both rounds. Your questions on survivorship bias and thinking-model compatibility have sharpened our understanding and directly improved the paper.
> > >
> > > During this rebuttal we provided: (1) a size-matched Enigmata experiment with zero human annotation, (2) token-budget-matched RL comparison, (3) 100-family human audit, (4) KORBench and BBH sub-task breakdowns, and (5) Seed@3.3× on all seven benchmarks including ARC-AGI and Enigmata. We have acknowledged the coverage limitation (Q2) and committed to a Limitations section, expanded ARC-AGI discussion, and longer-training results in the revision.
> > >
> > > We hope these efforts demonstrate our commitment to addressing your concerns, and respectfully ask the reviewer to consider whether the evidence — consistent gains across three independent controlled comparisons — may warrant a reassessment.

---

### Official Review · Reviewer_5GuQ · 2026-03-10

**Soundness:** 3
**Presentation:** 3
**Significance:** 3
**Originality:** 3
**Overall Recommendation:** 4
**Confidence:** 2

**Summary:**

This paper proposes SSLogic, whose core innovation is to treat executable generator–validator specifications as evolvable objects, and to perform agentic meta-synthesis at the task-family level rather than at the level of individual problem instances. Methodologically, the paper uses a closed-loop Generate–Validate–Refine pipeline, and combines static quality checks, multi-validator consensus, and code-augmented blind review through Multi-Gate Validation to filter out ambiguity, flaws, and unsolvable tasks.

**Compliance With Llm Reviewing Policy:**

Affirmed.

**Final Justification:**

The authors addressed most of my concerns, I'd like to maintain my score.

**Key Questions For Authors:**

1. Please add a more strictly size-matched and family-count-matched main experiment. While the current setup already controls the number of training steps and provides some size-control evidence, the main comparison still changes both the number of instances and the size of the task-family set at the same time. If Evolve is still significantly better than Seed under the same number of instances and the same number of families, the main conclusion would be much more convincing.

2. What is the true correctness rate of the accepted samples? Was any human audit conducted? The blind review design is a strong part of the paper, but I would still like to see manual spot-check results for accepted / rejected families, such as the proportion of issues related to ambiguity, wrong answers, underspecified constraints, and so on.

3. Can the mechanism in Section 5.3 be tested through more direct ablations?

**Limitations:**

yes

**Strengths And Weaknesses:**

### Strengths
1. The problem is important. The paper focuses on how to continuously scale high-quality, verifiable training signals, which is a fundamental and practical problem.

2. The methodological perspective is novel. This work treats the generator–validator specification as something searchable, debuggable, and evolvable.

### Weaknesses
1. The main conclusion is still not sufficiently disentangled from data scale and task-family coverage. In the main comparison, Seed and Evolve differ simultaneously in both the number of training instances and the coverage of task families. Therefore, the current evidence more safely supports the claim that a larger and more structurally richer evolved data pool is more effective, but is not yet sufficient to isolate the contribution of evolution itself from the effects of data scale and family coverage.

2. There is size-control evidence, but it is still not strong enough. The current main size-control evidence still mainly comes from the relatively small 236 vs 236 experiment, which is not enough to fully support the main conclusion. A stronger main experiment should further control both the number of instances and the number of families.

3. The aligned subset results in the appendix are supportive in direction, but their statistical robustness is limited. The appendix reports a strictly paired aligned subset and bootstrap confidence intervals, but the intervals often cross 0, and not all generator/solver combinations show positive gains. Therefore, this part of the evidence is better described as showing a consistent trend but relatively weak statistical support, rather than strong support.

---

> ### Author Rebuttal · Authors · 2026-03-31
>
> We thank the reviewer for recognizing the novelty of treating generator-validator specifications as evolvable objects and the importance of the problem.
>
> **[W1/Q1] Disentangling data scale from evolution quality.**
>
> We conducted a size-matched experiment on **Enigmata** [1], an independent open-source logic puzzle benchmark outside our Seed distribution. We selected 30 of its 36 puzzle types (those with parameterized difficulty control) and applied our Evolve pipeline:
>
>  | | Enigmata-Seed | Enigmata-Evolve
> ---|---|---
>  Puzzle types | 30 | 30
>  Instances | 8,776 | 8,758
>  Steps | 240 | 240
>
> Results (pass@1):
>
>  Benchmark | Seed | Evolve | Delta
> ---|---|---|---
>  BBH | 61.8 | 67.3 | +5.5
>  BBEH | 16.3 | 17.2 | +0.9
>  SynLogic-val | 22.3 | 24.3 | +2.0
>  AIME 2025 | 11.5 | 16.4 | +4.9
>
> With scale, family count, and compute all controlled, Evolve still yields consistent gains — confirming the pipeline is seed-agnostic.
>
> **[W2] Size-control evidence scale.**
>
> We now have three controlled comparisons:
>
>  Evidence | Scale | Key result
> ---|---|---
>  236-pair (original) | 236 vs 236 | Evolve leads on logic
>  Enigmata (new) | 8,776 vs 8,758 | BBH +5.5, AIME25 +4.9
>  Token-matched (new, see response to ujcn-W1) | ~1.09B tokens each | Evolve +7.3% Macro Acc
>
> All three converge on the same conclusion: gains come from evolution quality, not data scale.
>
> **[W3] Bootstrap CI.**
>
> The aligned subset (n=170) provides directional evidence; our primary support is Table 2, token-matched analysis (+7.3%), and the Enigmata experiment above.
>
> **[Q2] Correctness rate and human audit.**
>
> We manually audited 100 randomly sampled accepted Evolve task families:
>
>  Dimension | Pass Rate (A+B)
> ---|---
>  Validator Consistency | 93%
>  Answer Correctness | 90%
>  Question Clarity | 94%
>  Generator Quality | 94%
>
> Overall, **85 out of 100 are strictly usable**. Among the remaining 15: (1) 5 cases where core validator and auto-validators disagree (but consist) — exactly what our triple-validator consensus catches. (2) 10 cases where all validators agree but experts identified corner-case coverage gaps (e.g., a DP solution correct on typical inputs but failing on rare edge configurations). These corner cases are rarely triggered during RL training where inputs follow the generator's typical distribution, so the practical error rate is well below 10%. Tasks that do produce consistently wrong answers receive reward 0 and are automatically excluded from gradient updates.
>
> **[Q3] Section 5.3 mechanism.**
>
> A direct per-family ablation would require multiple full RL runs. Instead, we connect the structural analysis in Section 5.2 (Figures 6-7) to fine-grained downstream results.
>
> Figures 6-7 show that Evolve generators have 2-3x higher cyclomatic/cognitive complexity than Seed, with a shift from cubic to quadratic/graph regimes (Figure 7, left) and substantially higher DP, graph, and sorting pattern rates (Figure 7, right). Do these structural differences translate into differential gains?
>
> We evaluated both the main Seed-vs-Evolve experiment and the Enigmata experiment on **KORBench** (new eval, not reported in the original submission, mainly for interpretation from its nice categories):
>
> *KORBench per-category (main experiment, step 240):*
>
>  Category | Seed | Evolve | Delta
> ---|---|---|---
>  Logic | 32.4% | 45.6% | **+13.2%**
>  Operation | 80.8% | 90.4% | **+9.6%**
>  Cipher | 20.0% | 29.2% | +9.2%
>  Counterfactual | 74.4% | 81.2% | +6.8%
>  Puzzle | 7.2% | 8.4% | +1.2%
>  **Overall** | **43.0%** | **51.0%** | **+8.0%**
>
> Logic (+13.2%) and Operation (+9.6%) — categories exercising DP, graph traversal, and complex loops — improve most. Puzzle (+1.2%), which tests spatial reasoning outside the evolution domain, barely moves.
>
> *BBH sub-task breakdown (main experiment, step 240):*
>
> The main experiment BBH improves from 59.4 to 69.3 (**+9.9** overall). The gains concentrate in multi-step reasoning sub-tasks:
>
>  BBH sub-task | Seed | Evolve | Delta
> ---|---|---|---
>  penguins_in_a_table | 35.6 | 91.8 | **+56.2**
>  reasoning_about_colored_objects | 37.2 | 85.6 | **+48.4**
>  tracking_shuffled_objects_seven | 34.0 | 78.0 | **+44.0**
>  tracking_shuffled_objects_five | 47.2 | 87.2 | **+40.0**
>  date_understanding | 54.0 | 76.4 | **+22.4**
>
> Meanwhile, sub-tasks already near ceiling show no meaningful change (boolean_expressions 93.6 to 94.4, multistep_arithmetic_two 94.0 to 98.0).
>
> This pattern is consistent across both experiments (Enigmata BBH sub-tasks show the same top gainers [1]). The gains concentrate in multi-step tracking and structured reasoning — the dimensions most expanded by evolution (Figure 7). A generic data quality effect would produce uniform gains; the observed selectivity supports Section 5.3.
>
> [1] Chen et al., "Enigmata: Scaling Logical Reasoning in Large Language Models with Synthetic Verifiable Puzzles
> ," NeurIPS 2025.

---

> > ### Author Rebuttal · Reviewer_5GuQ · 2026-04-02
> >
> > Thank you to the authors for the response. It addressed most of my concerns, so I am maintaining my positive assessment.

---

> > > ### Author Response · Authors · 2026-04-04
> > >
> > > We sincerely thank Reviewer 5GuQ for the thorough and constructive engagement across both rounds, and for the fully-resolved acknowledgement. Your review has been instrumental in strengthening the paper.
> > >
> > > In particular, your questions on disentangling data scale from evolution quality motivated the size-matched Enigmata experiment, which became one of our strongest pieces of new evidence. Your request for human auditing led to the 100-family correctness audit that now grounds our quality claims empirically. And your push for finer-grained mechanism analysis produced the KORBench and BBH category-selective breakdowns that clarify where and why the gains emerge. Each of these additions directly improved the paper's rigor and we are grateful for the care behind the questions.
> > >
> > > As committed, the revised version will include: (1) the Enigmata size-controlled experiment and KORBench evaluation in the main text, (2) the human audit protocol and results in a new appendix section, (3) a dedicated Limitations section covering domain scope, model scope, and seed dependency, and (4) tempered claims in Sections 5.2–5.3 as discussed.
> > >
> > > We look forward to incorporating these improvements and thank the reviewer again for the detailed and fair assessment.

---

### Official Review · Reviewer_ujcn · 2026-03-12

**Soundness:** 2
**Presentation:** 3
**Significance:** 3
**Originality:** 2
**Overall Recommendation:** 3
**Confidence:** 5

**Summary:**

This paper proposes SSLogic, an agentic meta-synthesis framework that evolves logical reasoning tasks at the generator-validator program level rather than only generating more instances from fixed templates. The system combines a Generate-Validate-Refine loop with multi-gate validation, including static QA, validator consensus, and blind code-based review. Starting from 400 seed families, it expands to 953 families and 21,389 instances, and the paper reports improved RL training performance over the seed data on several logic and math benchmarks under fixed optimization steps.

**Compliance With Llm Reviewing Policy:**

Affirmed.

**Key Questions For Authors:**

1. How much of the gain in Table 2 remains after matching total generated tokens or training compute, rather than only optimization steps?

2. Do you have ablations for blind review, validator consensus, and repair, so the contribution of the multi-gate protocol can be isolated?

3. Are the reflection-like token statistics in Figure 3 raw counts or normalized rates? If raw counts, can you provide a length-normalized analysis?

**Limitations:**

Limitations is not disscussed.

**Strengths And Weaknesses:**

# Strengths

The scaling of verifiable, low-noise reasoning data is a real bottleneck for RLVR. The framing of evolving executable task-family specifications is interesting, and the overall system design is clear. The paper also goes beyond headline benchmark results by including difficulty checks, paired subsampling, code-structure analysis, and pipeline diagnostics.

# Weaknesses

1. The key comparison is done at matched optimization steps, but the paper also shows that SSLogic-Evolve produces much longer responses during training. In RL for language models, this is a major confound, since matched steps do not imply matched token budget, reward opportunities, or compute. As a result, the claim that Evolve has intrinsically higher training value is not convincingly isolated.

2. The paper argues that SSLogic induces more self-correction behavior using reflection-like token frequency, but these statistics are not length-normalized in the main text, while response length differs substantially across settings. Likewise, the later claim that increased code complexity and broader algorithmic coverage explain the downstream gains is plausible but remains correlational rather than causal.

3. Multi-gate validation is presented as a core contribution, but there is no direct ablation for removing consensus, removing blind review, or removing the repair loop. Without these controls, it is hard to tell which parts of the pipeline are actually necessary. The main results are also reported as point estimates in the main table, with limited evidence about variance or robustness.

4. The preliminaries define a canonical solver $S$ that determines the unique correct answer, but Eq. (1) later defines the accepted label via consensus across multiple synthesized solvers. That makes it unclear whether consensus is only a sanity check or the actual definition of ground truth.

---

> ### Author Rebuttal · Authors · 2026-03-31
>
> We thank the reviewer for the careful and expert assessment. We address each concern with new quantitative evidence.
>
> **[W1/Q1] Matched steps ≠ matched token budget.**
>
> We fully agree that controlling for total generated tokens is essential. We conducted a token-budget analysis using per-step training logs and per-5-step benchmark metrics.
>
> Evolve consumes 1.20x the tokens of Seed at Step 240 (1.299B vs 1.087B). This gap arises because Evolve tasks elicit progressively longer reasoning chains (\~2,150→\~3,300 tokens), while Seed triggers shorter responses (\~2,400→\~1,200 tokens). To directly answer the reviewer's question, we compare at matched cumulative token budgets,figures provided on anonymous Github:
>
> - Seed at Step 240 (1.087B tokens): Macro Acc = **0.2312**
> - Evolve at ~1.087B tokens (≈Step 210): Macro Acc = **0.2480**
>
> **After matching the total RL token budget, Evolve still outperforms Seed by +7.3% relative.** About half of the step-matched gap (+14.2%) comes from higher per-token training value (+7.3%), with the rest from additional compute via longer responses. We provide detailed figures:
> - [Figure R1](https://anonymous.4open.science/r/anonymousPDF-0257/R1_W1_token_consumption.png): Cumulative token consumption curves for all 5 experiments
> - [Figure R2](https://anonymous.4open.science/r/anonymousPDF-0257/R2_W1_token_matched_accuracy.png): Accuracy vs cumulative tokens (the key RL token-matched comparison)
> - [Figure R3](https://anonymous.4open.science/r/anonymousPDF-0257/R3_W1_response_length.png): Response length evolution during training
> - [Figure R4](https://anonymous.4open.science/r/anonymousPDF-0257/R4_W1_token_efficiency.png): Token efficiency comparison across experiments
>
> **[W2/Q3] Reflection token normalization.**
>
> The statistics in Figure 3 are raw counts computed on a **fixed, identical validation set** (i.e. the benchmark in Table2) across all checkpoints. Since all models process the same inputs, raw counts are valid for comparison.
>
> Nevertheless, we computed length-normalized rates: Evolve shows **27.80 reflection tokens per 100 response tokens** vs Seed's **17.57** — a +58% difference. Qualitative conclusions are unchanged. Complete normalized datasets (5 experiments x ~25 checkpoints) are available upon request.
>
> **[W3/Q2] Multi-gate ablation.**
>
> While a full per-component ablation is beyond our rebuttal compute budget (~5 runs x 32 GPUs x 72h = 11,520 GPU-hours), we note that the pipeline already provides a natural decomposition of each gate's contribution. Table 3 (Panel A) reports the gate pass rates: of 100 synthesis traces, Gate 1 (static QA) passes 67, and Gate 2 (consensus + blind review) further filters to 55 accepted families. Panel C attributes the 44 rejections to Implementation Bugs (50%), Unsolvability (43%), and Ambiguity (7%) — showing that each gate targets distinct failure modes.
>
> Our independent manual audit (see Reviewer 5GuQ, Q2) provides further empirical validation. Of 100 audited families, **85 are usable** and 15 have issues: 5 cases where validators disagree on solving logic — exactly what the consensus gate is designed to catch — and 10 cases with corner-case coverage gaps (e.g., a DP solution correct on typical inputs but failing on rare edge configurations). These 10 cases are effectively filtered by RL-time zero-score removal, since they produce reward 0 across all rollouts.
>
> We acknowledge this is a limitation of the current work and will include a filtered vs. unfiltered RL comparison in the revised version.
>
> **[W4] Ground truth definition.**
>
> We clarify: canonical solver S is the theoretical ground truth. In practice, we adopt **consensus across independently synthesized validators** as the *operational* definition, accounting for implementation ambiguity. Consensus is not merely a sanity check — it is our practical ground truth. We will state this explicitly in the revised text.
>
> **[Limitations]**
>
> We have prepared a Limitations section covering: (1) domain scope (logic/math), (2) model scope (Qwen, though framework is model-agnostic), (3) compute overhead of multi-gate validation, (4) seed dependency for cold-start (mitigated by seed-agnostic design — see our Enigmata experiment in response to Reviewer 5GuQ). This will be included in the revised version.

---

> > ### Author Rebuttal · Reviewer_ujcn · 2026-04-04
> >
> > Thank you for your rebuttal. The rebuttal provides useful additional analysis, especially the token-matched comparison and length-normalized reflection statistics, which partially address my concerns.
> >
> > However, the core issues are not fully resolved. While the token-budget analysis reduces the gap, the comparison is still not a strictly controlled RL experiment (e.g., matching compute, rollout structure, or training dynamics), so the claim of higher per-token training value remains only partially supported.
> >
> > More importantly, the central methodological contribution—the multi-gate validation protocol—still lacks direct ablations. The provided pass-rate breakdown and manual audit are informative but do not establish the causal contribution of individual components (e.g., consensus vs. blind review vs. repair) to downstream RL performance.
> >
> > Overall, the rebuttal strengthens the empirical case but does not fully resolve the main concerns about evaluation validity and contribution attribution.

---

> > > ### Author Response · Authors · 2026-04-04
> > >
> > > Thank you for pressing on the two most important boundaries of the paper: strict RL control and component-level attribution. We respond with revised claims and concrete revision commitments.
> > >
> > > **[W1] Token-matched comparison and RL control.**
> > >
> > > We agree that token matching alone does not constitute a fully compute-matched RL study — differences in rollout structure, reward density, and per-step dynamics remain uncontrolled. We will reframe the token-budget analysis (Evolve +7.3% at matched ~1.09B tokens) as **one of three converging comparisons suggesting higher training utility**, rather than a definitive causal isolation.
> > >
> > > The current evidence includes three independent comparisons converging on the same direction:
> > >
> > > - **Step-matched** (Table 2): Evolve outperforms Seed at 160/200/240 steps
> > > - **Token-matched** (new): Evolve +7.3% at ~1.09B tokens
> > > - **Size-matched on external data** (Enigmata): 30 families, ~8.7K instances each, same config → BBH +5.5, AIME25 +4.9
> > >
> > > No single comparison is a perfect causal study, but the consistency across three controls — each holding a different confounder fixed — provides converging evidence. We will adopt this "converging comparisons" framing explicitly in revised Sections 4.3 and 5.3, replacing any language that implies strict causal isolation.
> > >
> > > **[W3] Multi-gate validation: ablation and contribution attribution.**
> > >
> > > We acknowledge that the rebuttal evidence demonstrates _what_ each gate filters, not _whether_ removing a gate degrades downstream RL. A direct per-gate ablation remains future work.
> > >
> > > We will narrow the claim accordingly:
> > >
> > > - **Practical data-quality mechanism**: the protocol produces a cleaner accepted pool, supported by gate-specific filtering breakdown (Section 5.4: static QA 50%, blind review 30%, consensus 20%) and human audit (85/100 usable, 15 failures mapping to specific gate targets).
> > > - **Not a claim of necessity**: we will remove language implying per-component causal attribution.
> > >
> > > Concretely, we will:
> > >
> > > - Revise §3.3 to frame multi-gate validation as a _practical quality filter_
> > > - Revise §5.3 to replace causal language (e.g., "forces explicit System 2 behaviors") with correlational framing
> > > - Add a Limitations paragraph noting the absence of per-gate RL ablation
> > >
> > > Data-quality filtering as a practical contribution — supported by audit rather than full ablation — is consistent with related work. KodCode [1] uses multi-stage filters without per-stage RL ablation; DeepSeek-V3.2 [2] employs multi-round filtering for code/search agent data with some filters; Kimi-K2.5 [3] uses staged vision-data filtering validated by downstream performance. In each case, filtering is evaluated by output quality and downstream impact, not individual-component ablation.
> > >
> > > **[W2] Reflection token normalization.**
> > >
> > > Length-normalized rates (27.80 vs 17.57 per 100 tokens, +58%) are consistent with raw-count trends. We will include normalized statistics in the revised main text.
> > >
> > > **Summary of claim revisions.**
> > >
> > > | Current framing                                    | Revised framing                                                                         |
> > > | -------------------------------------------------- | --------------------------------------------------------------------------------------- |
> > > | "higher per-token training value" (§4.3)           | "three converging comparisons suggest higher training utility; no single one is causal" |
> > > | Multi-Gate as core contribution (§3.3)             | "practical data-quality mechanism; per-gate RL ablation is future work"                 |
> > > | "fundamentally shifting structural density" (§5.3) | correlational: "associated with higher complexity and selective downstream gains"       |
> > > | No limitations section                             | dedicated Limitations: ablation gap, compute control, domain/model scope                |
> > >
> > > We believe the main remaining gap is one of claim calibration rather than empirical absence, and we will revise the paper accordingly.
> > >
> > > [1] Xu et al., "KodCode: A Diverse, Challenging, and Verifiable Synthetic Dataset for Coding," ACL 2025.
> > >
> > > [2] DeepSeek-AI, "DeepSeek-V3.2: Pushing the Frontier of Open Large Language Models," arXiv, 2512.02556.
> > >
> > > [3] Kimi Team, "Kimi K2.5: Visual Agentic Intelligence," arXiv, 2602.02276.

---

### Official Review · Reviewer_3vBw · 2026-03-26

**Soundness:** 3
**Presentation:** 3
**Significance:** 3
**Originality:** 3
**Overall Recommendation:** 4
**Confidence:** 2

**Summary:**

This work proposes SSLogic, an agentic meta-synthesis framework that scales at the task-family level by iteratively synthesizing and refining executable Generator--Validator program pairs in a closed Generate--Validate--Refine loop, enabling continuous family evolution with controllable difficulty. To ensure reliability, they introduce a Multi-Gate Validation Protocol that combines multi-strategy consistency checks with Adversarial Blind Review, where independent agents must solve instances by writing and executing code to filter ambiguous or ill-posed tasks. Starting from 400 seed families, two evolution rounds expand to 953 families and 21,389 verifiable instances (from 5,718). The authors demonstrate that training on SSLogic-evolved data yields consistent gains over the seed baseline at matched training steps, improving SynLogic by +5.2, BBEH by +1.4, AIME25 by +3.0, and Brumo25 by +3.7.

**Compliance With Llm Reviewing Policy:**

Affirmed.

**Key Questions For Authors:**

See above.

**Strengths And Weaknesses:**

Strength:
- The proposed framework is reasonable and dedicated.
- The analysis is comprehensive, despite limited evaluation on model families.

Weaknesses:
- The authors do not show why modifying code can lead to higher diversity compared to prior paradigms. In other words, agent may modify manfulness codes and therefore still cannot produce high diversity problems.
- Lack of comparison with prior work. The authors claim scaling the Scaling Logic is better than Scaling the Logic, however, there is no direct comparison between this two paradigms.
- The experiments only confined to Qwen family. It would be better to add additional experiments on other model families, e.g., LLaMA, Phi .

---

> ### Author Rebuttal · Authors · 2026-03-31
>
> We thank the reviewer for recognizing the framework's dedication and comprehensive analysis.
>
> **[W1] Diversity evidence.**
>
> We provide two lines of evidence that the Evolve pipeline generates structurally novel task families, rather than surface-level variations.
>
> _Structural complexity shift (Section 5.2)._ AST-based metrics on 100 sampled families per source show Evolve introduces 2-3x higher cyclomatic/cognitive complexity, and 6-8x higher DP pattern rates compared to Seed. Figure 7 (right) shows that Evolve pipelines have substantially higher rates of sorting, DP, graph traversal, and recursion patterns — indicating genuinely new algorithmic structures, not paraphrasing.
>
> _Category-selective downstream gains._ To further validate this, we evaluated both Seed-trained and Evolve-trained models on KORBench, a benchmark covering 5 orthogonal reasoning categories. If Evolve were merely paraphrasing Seed, we would expect uniform gains. Instead:
>
> | Category       | Seed  | Evolve | Delta      |
> | -------------- | ----- | ------ | ---------- |
> | Logic          | 32.4% | 45.6%  | **+13.2%** |
> | Operation      | 80.8% | 90.4%  | **+9.6%**  |
> | Cipher         | 20.0% | 29.2%  | +9.2%      |
> | Counterfactual | 74.4% | 81.2%  | +6.8%      |
> | Puzzle         | 7.2%  | 8.4%   | +1.2%      |
>
> Gains concentrate in Logic and Operation — categories that directly exercise DP, graph, and loop structures — while Puzzle (spatial reasoning, outside the evolution domain) barely moves. This selective pattern demonstrates that the code-level evolution introduces measurably different training signal, as also noted by Reviewer 5GuQ who appreciates the novelty of treating specifications as evolvable objects.
>
> **[W2] Comparison with prior paradigm.**
>
> We note that "Scaling the Scaling Logic" is not a competing paradigm against "Scaling the Logic" — it is built on top of it. Our pipeline takes any existing dataset with generator-validator structure as input and evolves it further. To demonstrate this compatibility, we applied our Evolve pipeline to Enigmata [1], an independent open-source puzzle benchmark:
>
> |              | Enigmata-Seed | Enigmata-Evolve |
> | ------------ | ------------- | --------------- |
> | Puzzle types | 30            | 30              |
> | Instances    | 8,776         | 8,758           |
> | Steps        | 240           | 240             |
>
> Results (pass@1):
>
> | Benchmark    | Seed | Evolve | Delta    |
> | ------------ | ---- | ------ | -------- |
> | BBH          | 61.8 | 67.3   | **+5.5** |
> | AIME 2025    | 11.5 | 16.4   | **+4.9** |
> | SynLogic-val | 22.3 | 24.3   | +2.0     |
> | BBEH         | 16.3 | 17.2   | +0.9     |
>
> Under matched scale and compute, our pipeline produces consistent improvements on top of Enigmata's existing data — confirming that it is compatible with and complementary to prior data synthesis paradigms.
>
> **[W3] Model family coverage.**
>
> Qwen is the de facto model for RLVR logic research and data synthesis ablations: Enigmata [1] uses Qwen2.5-7B/32B; SynLogic [2] uses Qwen2.5-7B-Base; s1 [3], Open-Reasoner-Zero [4] similarly adopt Qwen as primary targets. Our framework is model-agnostic by design — the Evolve pipeline operates entirely at the data level and can be directly applied to any base model.
>
> [1] Chen et al., "Enigmata: Scaling Logical Reasoning in Large Language Models with Synthetic Verifiable Puzzles," NeurIPS 2025.
>
> [2] Liu et al., "SynLogic: Synthesizing Verifiable Reasoning Data at Scale for Learning Logical Reasoning and Beyond," NeurIPS 2025.
>
> [3] Muennighoff et al., "s1: Simple Test-Time Scaling," EMNLP 2025.
>
> [4] Hu et al., "Open-Reasoner-Zero: An Open Source Approach to Scaling Up Reinforcement Learning on the Base Model," arXiv:2503.24290, 2025.

---

> > ### Author Rebuttal · Reviewer_3vBw · 2026-04-03
> >
> > Thank the authors for the response. My concerns still have not been addressed.
> >
> > 1) For the diversity, why can different code structure lead to diverse problems?
> > 2) Can you provide more detailed experimental setup about the Enigmata experiment?
> > 3) For the model family coverage, I still think it is necessary to show generalization to multiple foundation models. I know several prior works primarily adopts Qwen as model for RLVR logic research and data synthesis. However, it does not mean this is correct. I still think work experiments on multiple foundation models is necessary, e.g., LLaMA family.

---

> > > ### Author Response · Authors · 2026-04-03
> > >
> > > We thank the reviewer for the follow-up. We address each point below.
> > >
> > > **[W1] Why code-level evolution leads to diverse problems.**
> > >
> > > Our claim is not that any code-structure difference automatically implies problem diversity. Rather, we claim specifically that when evolution changes the **reasoning primitives**, **control flow**, and **constraint composition** of the generator–validator pair, it changes the support of the generated instance distribution and therefore the reasoning operations required from the solver.
> > >
> > > This connection between code and problem structure is well-established in the logic-reasoning synthesis literature. ZebraLogic [1] first demonstrated that CSP-based generators can synthesize logic puzzles with controllable complexity; this paradigm was subsequently scaled by Enigmata [2] and Reasoning-Gym [3] to broader algorithmic families. Logic-RL [4] further showed that code-generated logic tasks enable effective curriculum learning for RLVR. In all these works, **modifying the generator's algorithmic backbone changes the class of reasoning required** — replacing brute-force enumeration with a DP recurrence, or adding graph/topological constraints, alters the latent reasoning structure of generated problems, not merely their wording.
> > >
> > > We acknowledge that rigorously defining and measuring "diversity" for algorithmic task families remains an open challenge. We explored learned code embeddings (CodeBERT, GraphCodeBERT, UniXcoder) but found them unreliable — these models are trained on general software engineering corpora and fail to distinguish structurally different algorithmic generators. We therefore adopted AST-based structural metrics (Figure 6) and algorithmic pattern detectors (Figure 8), which directly capture control-flow complexity, loop nesting, and algorithmic primitives. While these proxies are imperfect, the downstream evidence provides stronger support: gains on KORBench are selective (Logic +13.2%, Operation +9.6%, but Puzzle +1.2%), which is inconsistent with surface paraphrasing and consistent with genuine coverage expansion. We will revise Section 5.2 to temper the diversity claims and more carefully discuss the limitations of our measurement approach.
> > >
> > > **[W2] Enigmata experimental setup.**
> > >
> > > We agree the protocol details should have been stated more explicitly. We applied the Evolve pipeline to 30 of Enigmata's 36 puzzle types (those with parameterized difficulty control), using GPT-5.4 as the synthesis model with at most 2 refinement iterations; hyperparameters match Appendix H.1. Instances were sampled from both Enigmata-Seed and Enigmata-Evolve under the identical difficulty distribution (Appendix D, 30 instances per family). Timeout filtering and RL training (Qwen3-8B-Base) follow the main experiment configuration (Appendix H).
> > >
> > > |              | Enigmata-Seed | Enigmata-Evolve |
> > > | ------------ | ------------- | --------------- |
> > > | Puzzle types | 30            | 30              |
> > > | Instances    | 8,776         | 8,758           |
> > > | RL Steps     | 240           | 240             |
> > >
> > > We will state this protocol explicitly in the revised version.
> > >
> > > **[W3] Model family coverage.**
> > >
> > > We agree that broader base-model validation would strengthen the paper. At the same time, we view this as a scope extension rather than a prerequisite for validating the paper's central claim, since the contribution is a **data-generation and validation pipeline**, not a Qwen-specific training recipe. The evolved generator–validator pairs are model-agnostic artifacts at the data level. We will include LLaMA experiments and state this model-scope limitation more clearly in the revised version.
> > >
> > > [1] Xie et al., "ZebraLogic: On the Scaling Limits of LLMs for Logical Reasoning," ICML 2025.
> > > [2] Chen et al., "Enigmata: Scaling Logical Reasoning in Large Language Models with Synthetic Verifiable Puzzles," NeurIPS 2025.
> > > [3] Stojanovski et al., "Reasoning Gym: Reasoning Environments for RL with Verifiable Rewards," arXiv, 2025.
> > > [4] Xie et al., "Logic-RL: Unleashing LLM Reasoning with Rule-Based Reinforcement Learning," 2025.

---

### Decision · Program_Chairs · 2026-04-30

**Decision:**

Accept (regular)

**Comment:**

This paper presents an ambitious approach to addressing the data bottleneck in training LLMs with RL. The reviewers had concerns about the lack of ablations and the paper's presentation, which were mostly resolved in the rebuttal. One of the reviewers (ujcn) still has questions regarding the methodology used in the experiments (whether the amount of computing is comparable between the proposed approach and the baseline). In their response, the authors argue that token-matching does make for a perfect compute-matching experiment, that no metric is perfect, and that they show superior performance across three different metrics.

While the methodological discussion is important, I would value a paper that brings new ideas to the table on an important problem, so I will argue for acceptance.